# PROXI: Challenging the GNNs for Link Prediction

**Astrit Tola**                                                                 *astrit.tola@utdallas.edu*
*Department of Mathematical Sciences*
*University of Texas at Dallas*

**Jack Alec Myrick**                                                            *jack.myrick@utdallas.edu*
*Department of Computer Science*
*University of Texas at Dallas*

**Baris Coskunuzer**                                                           *coskunuz@utdallas.edu*
*Department of Mathematical Sciences*
*University of Texas at Dallas*

**Reviewed on OpenReview:** *https://openreview.net/forum?id=u9EHndbiVw*

## Abstract

Over the past decade, Graph Neural Networks (GNNs) have transformed graph representation learning. In the widely adopted message-passing GNN framework, nodes refine their representations by aggregating information from neighboring nodes iteratively. While GNNs excel in various domains, recent theoretical studies have raised concerns about their capabilities. GNNs aim to address various graph-related tasks by utilizing such node representations, however, this one-size-fits-all approach proves suboptimal for diverse tasks.

Motivated by these observations, we conduct empirical tests to compare the performance of current GNN models with more conventional and direct methods in link prediction tasks. Introducing our model, PROXI, which leverages proximity information of node pairs in both graph and attribute spaces, we find that standard machine learning (ML) models perform competitively, even outperforming cutting-edge GNN models when applied to these proximity metrics derived from node neighborhoods and attributes. This holds true across both homophilic and heterophilic networks, as well as small and large benchmark datasets, including those from the Open Graph Benchmark (OGB). Moreover, we show that augmenting traditional GNNs with PROXI significantly boosts their link prediction performance. Our empirical findings corroborate the previously mentioned theoretical observations and imply that there exists ample room for enhancement in current GNN models to reach their potential. Our code is available at `https://github.com/workrep20232/PROXI`

## 1  Introduction

In an era characterized by the complex web of digital connections, understanding and predicting the formation of links between entities in complex networks has become a crucial challenge. Whether it's predicting social connections in online social networks, anticipating collaborations between researchers, or forecasting potential interactions in recommendation systems, link prediction has emerged as a fundamental task in graph representation learning. With the ongoing evolution of our societies and technologies, the significance of link prediction amplifies, considering its capability to refine a broad spectrum of applications—ranging from personalized content suggestions to strategically targeted marketing.

Over the past decade, Graph Neural Networks (GNNs) have achieved a significant breakthrough in graph representation learning (Wu et al., 2020). Despite their consistently superior performance compared to state-of-the-art (SOTA) results, seminal papers like Xu et al. (2019); Morris et al. (2019); Li & Leskovec (2022) have shown that the expressive capacity of message-passing GNN (MP-GNNs) models is not better

than decades-old Weisfeiler-Lehman algorithm. Furthermore, Loukas (2020a;b); Sato (2020); Barceló et al. (2020) theoretically showed that MP-GNNs' representational power is limited. These theoretical discoveries suggest that existing GNN models may not be fully leveraging their potential to integrate information from local neighborhoods and domain-specific knowledge when learning node embeddings, a fundamental process in the creation of node representations.

GNNs face another challenge with their one-size-fits-all approach. They employ a range of methods and architectural designs to generate robust node embeddings, merging neighborhood information with node attributes. These embeddings are then *tailored* to different graph representation learning tasks such as node classification, graph classification, or link prediction by adjusting the prediction head and loss function. While this approach is straightforward for node classification, it is highly indirect for graph classification and link prediction tasks (Zhu et al., 2024; Zhang et al., 2021). In link prediction, for instance, understanding the "proximity" between given node pairs is crucial to determining whether they will form a link. Common GNN algorithms achieve this by first learning node representations, $\mathbf{h}_u$ and $\mathbf{h}_v$, in a latent space $\mathbb{R}^m$, and then measuring "the distance" between these representations, e.g., $\mathbf{h}_u \cdot \mathbf{h}_v$. However, this method is not practically viable for link prediction. One reason is that distance is a transitive operation, while link formation is not. For example, having links between node pairs $u \sim v$ and $v \sim w$ does not guarantee a link between $u \sim w$. Yet, the proximity of $\mathbf{h}_u \sim \mathbf{h}_v$ and $\mathbf{h}_v \sim \mathbf{h}_w$ implies the proximity of $\mathbf{h}_u \sim \mathbf{h}_w$ due to the triangle inequality (Section 3.3). Hence, a more suitable approach for GNNs would be to directly learn the prediction heads for the tasks (e.g., node pair representation), bypassing the reliance on individual node embeddings.

In graph representation learning, *homophily* refers to the tendency of nodes with similar features or labels to be connected, reflecting the principle of *like attracts like.* In contrast, *heterophily* describes networks where nodes with dissimilar features or labels are more likely to connect, highlighting diversity in connections (See Section 3.1). Most GNNs, however, are designed based on the homophily assumption, where edges typically link nodes with similar attributes or labels, as commonly observed in citation networks. This homophilic bias poses a significant limitation in real-world applications, where heterophilic behavior is prevalent, such as in protein interaction and web networks. In these settings, traditional GNN models (Hamilton et al., 2017; Klicpera et al., 2019) often suffer from substantial performance degradation as shown by recent studies (Zhou et al., 2022; Zheng et al., 2022; Pan et al., 2021; Zhu et al., 2020; Luan et al., 2022).

In this paper, motivated by these, we aimed to test the performance of GNNs against conventional methods in link prediction tasks. First, we introduce a simple yet very direct ML model: *PROXI*, which combines all relevant proximity information about node pairs. We approach the link prediction problem simply as a binary classification, whether a node pair will form a link or not. Our method is a fusion of two types of proximity information: *structural proximity*, capturing the proximity of the node pair within the graph structure, and *domain proximity*, measuring their similarity within the attribute space. These indices provide direct embeddings of node pairs encoding the proximity information. Through the integration of these spatial and domain proximities with conventional ML methods, our models demonstrate highly competitive results compared to state-of-the-art GNNs across a diverse range of datasets, spanning both homophilic and heterophilic settings. We further show that integration of our PROXI model significantly enhances the performance of conventional GNN models in link prediction tasks.

**Our contributions:**

◇ We propose a scalable ML model *PROXI* for link prediction task, by adeptly merging the local neighborhood information and domain-specific node attributes.

◇ With only 20 indices, our model consistently achieves highly competitive results with SOTA GNN models in benchmark datasets for both homophilic and heterophilic settings. Hence, our simple model provides a critical baseline for new GNNs in link prediction.

◇ Our PROXI indices can easily be integrated with existing GNN models, leading to significant performance improvements up to 11%.

◇ Our results support recent theoretical studies, indicating that current GNNs may not be significantly better than traditional models. This underscores the need for novel approaches to unlock the full capabilities of GNNs.

## 2 Related Work

### 2.1 GNNs for Link Prediction

Much of the recent work on link prediction has been using GNNs. Most GNNs follow the message-passing framework, in which a node's representation is learned through an aggregation operation, to pool local neighborhood attributes, and an update operation, which is a learned transformation (Guo et al., 2023). Another common framework is the encoder-decoder framework in which the encoder learns node representations and the decoder predicts the probability of a link between two nodes (Guo et al., 2023). There are four main groups of GNNs currently used (Wu et al., 2020): recurrent GNNs (RecGNN) (Dai et al., 2018), convolutional GNNs (ConvGNNs) (Chiang et al., 2019), graph autoencoders (GAEs) (Bojchevski et al., 2018), and spatial–temporal GNNs (STGNNs) (Guo et al., 2019).

In the link prediction task, GNNs have shown outstanding performance in the past decade (Zhang, 2022; Zhu et al., 2021; Wang et al., 2023; Srinivasan & Ribeiro, 2020; Rossi et al., 2021). Zhao et al. (2022) use counterfactual links as a data-augmentation method to obtain robust and high-performing GNN models. In Yun et al. (2021), the authors applied novel approaches to improve learning structural information from graphs. In Zhu et al. (2021), the authors integrate the Bellman-Ford algorithm for path representations to their GNN model and obtain competitive results in both inductive and transductive settings. Wu et al. (2021) proposed an effective similarity computation method by employing a hashing technique to boost the performance of GNNs in link prediction tasks. Liu et al. (2022) developed high-performing GNNs for dynamic interaction graph setting. There is an overwhelming literature on GNNs for link prediction in the past few years, and a comprehensive review of these developments can be found in Zhang (2022); Liu et al. (2023); Wu et al. (2022).

Note that the majority of the GNN models above rely on the homophily assumption and demonstrate suboptimal performance in heterophilic networks (Zhu et al., 2020). Therefore, over the recent years, several efforts have been made to create GNNs that perform well in heterophilic networks (Zhou et al., 2022; Zheng et al., 2022; Pan et al., 2021; Luan et al., 2022).

Although this paper focuses on the link prediction task, we note that recent studies have demonstrated the competitive performance of traditional methods compared to GNNs in graph classification (Loiseaux et al., 2024; Chen et al., 2024) and node classification (Uddin et al., 2024; Luan et al., 2023a). These findings highlight significant opportunities for advancing GNNs through innovative approaches.

### 2.2 Proximity-Based Methods for Link Prediction

Before the GNNs, many of the common machine learning models often relied on feature engineering methods as a primary approach (Kumar et al., 2020; Menon & Elkan, 2011). One of the simplest approaches to link prediction is through proximity-based methods (Lü & Zhou, 2011), including local proximity indices (Wu et al., 2016), global proximity indices (Jeh & Widom, 2002), quasi-local indices (Liu & Lü, 2010), the local path index (Lü et al., 2009) and its extension (Aziz et al., 2020), and the Katz index (Katz, 1953; Coşkun et al., 2021) . In this case, the graphs are mostly assumed to be homophilic, and more similar nodes are deemed more likely to have a link.

Most of the former proximity metrics for link prediction can be categorized as local proximity indices. Let $S(u,v)$ denote a similarity score between two nodes $u$ and $v$, let $\mathcal{N}(u)$ denote the set of neighbors of $u$, and let $k_u$ denote the degree of $u$.

⋄ *# Common Neighbors* is the size of the intersection between two nodes' neighbors (Newman, 2001). This is equivalent to the number of paths of length 2 between two nodes. More common neighbors indicate a higher likelihood for a link. $\quad \mathcal{CN}(u,v) = |\mathcal{N}(u) \cap \mathcal{N}(v)|$

⋄ *Jaccard Coefficient* is a normalized Common Neighbor score (Jaccard, 1901), i.e., the probability of selecting a common neighbor of two nodes from all neighbors of those nodes. $\quad \mathcal{J}(u,v) = \frac{|\mathcal{N}(u) \cap \mathcal{N}(v)|}{|\mathcal{N}(u) \cup \mathcal{N}(v)|}$

⋄ *Salton Index* (cosine similarity) measures similarity using direction rather than magnitude (Singhal et al., 2001). $\quad \mathcal{S}_a(u,v) = \frac{|\mathcal{N}(u) \cap \mathcal{N}(v)|}{\sqrt{k_u k_v}}$

⋄ *Sørensen Index* was developed for ecological data samples (Sørensen, 1948), and it is more robust than Jaccard against outliers (McCune & Grace, 2002). $\mathcal{S}_o(u,v) = \frac{2|\mathcal{N}(u) \cap \mathcal{N}(v)|}{k_u + k_v}$

⋄ *Adamic Adar Index* measures the number of common neighbors between two nodes weighted by the inverse logarithm of their degrees (Adamic & Adar, 2003). It is defined as $\mathcal{AA}(u,v) = \sum_{w \in \mathcal{N}(u) \cap \mathcal{N}(v)} \frac{1}{\log|k_w|}$

We use these similarity metrics as our *structural proximity indices* for node pairs. We then effectively combine them with our *domain proximity indices*, which is the relevant similarity measure between the node attributes, to complete our proximity indices for our ML model.

## 3 Methodology

### 3.1 Problem Statement

Link prediction problems in graph representation learning can be categorized into different types based on the nature of the problem and the availability of information during the prediction process. Three common types are *transductive*, *inductive*, and *semi-inductive* link prediction.

Let $\mathcal{G} = (\mathcal{V}, \mathcal{E}, \mathcal{X})$ be a graph, where $\mathcal{V} = \{v_1, v_2, \ldots, v_n\}$ represents the set of nodes, $\mathcal{E} \subset \mathcal{V} \times \mathcal{V}$ represents the set of edges and $\mathcal{X}$ represents the attribute feature matrix ($n \times m$ size) where $\mathbf{X}_i \in \mathbb{R}^m$ is the $m$-dimensional attribute feature vector of node $v_i$. For the sake of simplicity, we focus on unweighted, undirected graphs, but our setup can easily be adapted to more general settings.

The main difference between these types comes from the availability of the information during the prediction process. We split the vertex and edge sets as *observed* (old) and *unobserved* (new) subsets, i.e. $\mathcal{V} = \mathcal{V}_o \cup \mathcal{V}_u$ and $\mathcal{E} = \mathcal{E}_o \cup \mathcal{E}_u$. Hence, in the training process, we are provided $\mathcal{G}_o = (\mathcal{V}_o, \mathcal{E}_o)$ information, and we are asked to predict the existence of a link in $\mathcal{E}_u$ for a given node pair in $\mathcal{V}$. However, the type of the problem is determined with respect to which subsets (i.e., $\mathcal{V}_o$ or $\mathcal{V}_u$) these node pairs are chosen from:

- *Transductive Setting:* Predict whether $e_{ij} \in \mathcal{E}_u$ where $v_i, v_j \in \mathcal{V}_o$.

- *Semi-inductive Setting:* Predict whether $e_{ij} \in \mathcal{E}_u$ where $v_i, v_j \in \mathcal{V}_o \cup \mathcal{V}_u$.

- *Inductive Setting:* Predict whether $e_{ij} \in \mathcal{E}_u$ where $v_i, v_j \in \mathcal{V}_u$. No local structure information is provided, only attribute vectors $\{\mathbf{X}_i\}$ are provided for $v_i \in \mathcal{V}_u$.

While, in the literature, the most common type is transductive setting, depending on the domain, the relevant question can come in any of these forms. To maintain focus in this paper, we align with the prevalent transductive setting, consistent with most contemporary GNN models. It is important to note, however, that our proposed ML model exhibits a high degree of versatility and can easily adapt to any of these settings.

**Heterophily.** Before presenting our model, we would like to recall the concepts of homophily and heterophily. Informally, homophily describes the tendency of edges in a graph to connect nodes that are similar, while heterophily describes the opposite tendency. Formally, homophily/heterophily is determined by the homophily ratio as follows: Let $\mathcal{G} = (\mathcal{V}, \mathcal{E})$ be a graph with $\mathcal{C} : \mathcal{V} \to \{1, 2, \ldots, N\}$ representing node classes. For each node $v$, let $\eta(v)$ be the number of adjacent nodes with the same class, and let $deg(v)$ denote the degree of node $v$. Then, the *node homophily ratio* of $\mathcal{G}$ is defined as: $H_n(\mathcal{G}) = \frac{1}{|\mathcal{V}|} \sum_{v \in \mathcal{V}} \frac{\eta(v)}{deg(v)}$. By definition, $H_n(\mathcal{G}) \in [0,1]$ for any graph $\mathcal{G}$. A common convention is that a graph $\mathcal{G}$ with $H_n(\mathcal{G}) \geq 0.5$ is called *homophilic*, and otherwise *heterophilic*. Another important homophily metric especially for link prediction task is the *edge homophily ratio*, which is the proportion of edges connecting nodes in the same class to all edges in the graph. i.e., $H_e(\mathcal{G}) = \frac{|\widetilde{\mathcal{E}}|}{|\mathcal{E}|}$ where $\widetilde{\mathcal{E}}$ represent the edges connecting same class nodes. Recently, various new metrics were introduced to study the homophily in graphs (Luan et al., 2023b; Zhu et al., 2020; Jin et al., 2022).

### 3.2 PROXI for Link Prediction

In the following, we give the details of our simple proximity-based model, PROXI. Our primary aim is to furnish our ML classifier with a comprehensive set of relevant and potentially valuable information about node pairs. In the domain of graph representation learning, most configurations offer two types of information about nodes. The first concerns the overall graph structure, providing spatial information about their local neighborhoods. The second involves node attributes derived from the specific problem domain, e.g. keywords of a paper in citation networks.

Essentially, the link prediction problem boils down to determining whether a node pair $(u, v)$ is poised to initiate an interaction or not. Therefore, their structural proximity and domain proximity (shared interests) play key roles in this decision-making process. In our model, we intentionally opt to allow the classifier the discretion to determine which indices to leverage, depending on the dataset and the setting (e.g., homophilic or heterophilic) at hand. While existing literature typically employs these indices individually or in pairs, our intuition leads us to believe that by aggregating all of this information, the ML classifier can make finer determinations within the latent space. Furthermore, these indices can synergistically reinforce each other, resulting in a more robust and accurate model.

In this context, our indices can be categorized into two distinct types: *Structural Indices* and *Domain Indices.* Structural indices draw upon the inherent graph structure and neighborhood information, while domain indices leverage proximity measures derived from the node attributes provided. We note that some of our indices have been integrated in earlier GNN modes like BUDDY and Labeling Trick, where we discuss them in Appendix B.1.

**Novel Structural Proximity Indices for Node Pairs** $(u, v)$. Within our PROXI model, we incorporate a range of structural indices. While a subset of these indices corresponds to established similarity indices as detailed in Section 2.2, along with their generalizations, we also introduce novel proximity indices. These newly defined indices are designed to capture finer insights from the local neighborhoods of node pairs.

The established indices are the Jacard, Salton, Sørensen, and Adamic Adar indices (Section 2.2). These are known to capture triadic closure information in networks, which is a key signature for link prediction problem (Huang et al., 2015). Furthermore, for Jacard, Salton, and Sørensen indices, we use their natural generalizations for 3-neighborhood versions as well as new indices like length-k paths and distance index.

$\diamond$ *# Length-k paths:* For a given $u, v \in \mathcal{V}$, we define *length-k paths index* $\mathcal{L}_k(u, v)$ as the total number of length-$k$ paths between the nodes $u$ and $v$. Notice that length-2 paths index is the same with common neighbors index, i.e.,

$$\mathcal{L}_2(u, v) = \mathcal{CM}(u, v) = |\mathcal{N}(u) \cap \mathcal{N}(v)|$$

$\diamond$ *Graph Distance:* For a given $u, v \in \mathcal{V}$, we define *distance index* $\mathcal{D}(u, v)$ as the length of the shortest path between $u$ and $v$ in $\mathcal{G}$. Note that when computing $\mathcal{D}(u, v)$, we remove the edge between $u$ and $v$ from the graph if $u$ and $v$ are adjacent nodes. Therefore, $\mathcal{D}(u, v) \geq 2$ for any $u \neq v \in \mathcal{V}$. The main motivation to define this index in this particular way is that in the test set, a priori, there won't be an edge between the nodes. Therefore, during the training ML classifier, this distance index provides valuable information to the ML classifier to distinguish positive and negative edges when combined with other indices.

$\diamond$ *3-Jaccard Index:* $\mathcal{J}^3(u, v)$ is a natural generalization of the original Jaccard Index by using $\mathcal{L}_3(u, v)$ the length 3-paths instead of $\mathcal{L}_2(u, v)$ length-2 paths.

$$\mathcal{J}^3(u, v) = \frac{\mathcal{L}_3(u, v)}{|\mathcal{N}(u) \cup \mathcal{N}(v)|}$$

By using a similar idea, we implement a comparable adaptation to Salton and Sørensen indices:

$\diamond$ *3-Salton Index:* By generalizing original Salton index to 3-neighborhoods, we introduce 3-Salton index:

$$\mathcal{S}_a^3(u, v) = \frac{\mathcal{L}_3(u, v)}{\sqrt{k_u k_v}}$$

$\diamond$ *3-Sørensen Index:* Similarly, by generalizing original Sørensen index to 3-neighborhoods, we introduce 3-Sørensen index:

$$\mathcal{S}_o^3(u, v) = \frac{2\mathcal{L}_3(u, v)}{k_u + k_v}$$

Hence, for a node pair $u, v \in \mathcal{V}$, we produce 10 structural PROXI indices as follows $\mathcal{J}(u,v), \mathcal{S}_a(u,v), \mathcal{S}_o(u,v), \mathcal{J}^3(u,v), \mathcal{S}_a^3(u,v), \mathcal{S}_o^3(u,v), \mathcal{AA}(u,v), \mathcal{L}_2(u,v), \mathcal{L}_3(u,v)$ and $\mathcal{D}(u,v)$.

**Novel Domain Proximity Indices for Node Pairs** $(u,v)$. Next, we describe our domain indices. Contrary to our structural indices, our domain indices do not use graph structure, but only the node attribute vectors. For a given graph with node attributes $\mathcal{G} = (\mathcal{V}, \mathcal{E}, \mathcal{X})$, let $\mathbf{X}_u \in \mathbb{R}^m$ represent the attribute vector for the node $u \in \mathcal{V}$. In the following, for a given node pair $u, v \in \mathcal{V}$, we extract our domain indices $\alpha(u,v)$ by using the similarity/dissimilarity of these node attribute vectors $\mathbf{X}_u$ and $\mathbf{X}_v$.

While our overarching argument to produce the domain proximity indices is the same, we adapt our approach to accommodate various formats of node attribute vectors $\{\mathcal{X}_u\}$. These formats can generally be categorized as follows.

**i. $\mathbf{X}_u$ is binary vector: All entries $\mathbf{X}_u^i \in \{0,1\}$**

In this case, we naturally interpret this as every binary digit in the vector $\mathbf{X}_u$ represents the existence or nonexistence of a property. For example, if $\mathcal{G}$ represents a citation network, where nodes represent papers, $\mathbf{X}_u$ can be a binary vector representing the existence or nonexistence of previously chosen keywords in the paper $u$. We define two similarity measures between $\mathbf{X}_u$ and $\mathbf{X}_v$.

⋄ *Common Digits:* If $\mathbf{X}_u$ is a binary vector, we define our *Common Digits* domain index $\mathcal{CD}(u,v)$ as the number of matching "1"s in the vectors $\mathbf{X}_u$ and $\mathbf{X}_v$. i.e., $\mathcal{CD}(u,v) = \#\{i \mid \mathbf{X}_u^i = \mathbf{X}_v^i = 1\}$. similarly, one can define an analogous index as the number of common "0"s to emphasize the common absent properties.

⋄ *Normalized Common Digits:* This domain index is a slight variation of the previous one with some normalization factor. In particular, if $\mathbf{X}_u$ and $\mathbf{X}_v$ have only a few positive digits in their vectors, having an equal number of common digits would result in them being considered more similar, in contrast to node pairs with numerous positive digits. We normalize this index by dividing it by the total number of positive digits in both vectors $\mathbf{X}_u$ and $\mathbf{X}_v$ (not counting the common positive digits twice). $\widehat{\mathcal{CD}}(u,v) = \frac{\#\{i \mid \mathbf{X}_u^i = \mathbf{X}_v^i = 1\}}{\#\{j \mid (\mathbf{X}_u + \mathbf{X}_v)^j \geq 1\}}$ Notice that the vector $\mathbf{X}_u + \mathbf{X}_v$ would have only $0, 1$, and $2$ digits where $2$s represent the common positive digits in $\mathbf{X}_u$ and $\mathbf{X}_v$. i.e., $\mathcal{CD}(u,v) = \#\{j \mid (\mathbf{X}_u + \mathbf{X}_v)^j = 2\}$

**ii. $\mathbf{X}_u^i$ takes finitely many values: $\mathbf{X}_u^i \in \{1, 2, \ldots, m\}$**

In this case, we make the assumption that a particular entry $\mathbf{X}_u^i$ of the vector $\mathbf{X}_u$ can assume a finite set of distinct values, such as $\mathbf{X}_u^i \in 1, 2, \ldots, m$. In such cases, we interpret this specific entry as representing some sort of "node class information" of a particular node attribute of $u$. Notably, if node classes are provided in the data, we take this information into account within this context. For instance, in citation networks, this data could correspond to the academic field of the paper (e.g., Mathematics, Computer Science, History) as an attribute of the node. Within this category, we establish two distinct domain indices.

⋄ *Class Identifier:* $\mathcal{CI}(u,v)$ is an $m$-dimensional binary vector to identify the classes of $u$ and $v$. i.e., if $\mathbf{X}_u^i = s$ and if $\mathbf{X}_v^i = t$, then we define $\mathcal{CI}(u,v)$ as an $m$-dimensional binary vector with all entries are 0 except $s^{th}$ and $t^{th}$ entries, which are marked 1. If $u$ and $v$ belong to the same class ($s = t$), then we $\mathcal{CI}(u,v)$ is all zeros except $s^{th}$ entry.

For example, if there are 5 classes ($m = 5$), and $\mathbf{X}_u = 2$ and $\mathbf{X}_v = 4$, we have $\mathcal{CI}(u,v) = [0\ 1\ 0\ 1\ 0]$. If $u$ and $v$ are in the same class, say $\mathbf{X}_u = \mathbf{X}_v = 3$, then we have $\mathcal{CI}(u,v) = [0\ 0\ 1\ 0\ 0]$.

⋄ *Common Class:* $\mathcal{MC}(u,v)$ is a one-dimensional binary index that is one if $s$ and $t$ are equal and zero otherwise.

**iii. $\mathbf{X}_u$ is a real-valued vector: $\mathbf{X}_u \in \mathbb{R}^m$**

When $\mathbf{X}_u$ is represented as a real-valued vector, it inherently serves as a node embedding within the attribute space $\mathbb{R}^m$. Therefore, the similarity or dissimilarity between two nodes is intuitively associated with the distance between these embeddings. We employ two distinct types of distance measurements as domain indices.

$\diamond$ $L^1$-*Distance:* Simply, we use $L^1$-norm (Manhattan metric) in the attribute space $\mathbb{R}^m$. If $\mathbf{X}_u = [a_1\ a_2\ \ldots\ a_m]$ and $\mathbf{X}_v = [b_1\ b_2\ \ldots\ b_m]$, we define

$$\mathfrak{D}_1(u,v) = \mathbf{d}(\mathbf{X}_u, \mathbf{X}_v) = \sum_{i=1}^{m} |a_i - b_i|$$

$\diamond$ *Cosine Distance:* Another popular distance formula using some normalization is the cosine distance/similarity. We define our cosine distance index as

$$\mathfrak{D}^{\mathfrak{c}}(u,v) = \frac{\mathbf{X}_u \cdot \mathbf{X}_v}{\|\mathbf{X}_u\| . \|\mathbf{X}_v\|}$$

Note that the node attributes may result from a combination of these three categories. In such instances, we incorporate all of them by dissecting the node attribute vector based on their respective subtypes and acquiring the corresponding domain attributes for the node pairs.

### 3.3 Non-Transitivity of Link Prediction Problem

In this part, we will *speculate* about the challenges associated with using node representations for link prediction tasks. After generating node representations $\{\mathbf{h}_u\}$, the goal of the prediction head is to leverage these embeddings to determine whether a given pair of nodes should form a positive or negative edge. Typically, the prediction head employs a similarity measure between node representations, implying that if two nodes have similar representations, they should form a link $((u,v) \to "+")$; otherwise, they should not $((u,v) \to "-")$. However, most similarity measures inherently adhere to some form of triangle inequality, such as cosine similarity.

$$\cos(\mathbf{h}_u, \mathbf{h}_w) \geq \cos(\mathbf{h}_u, \mathbf{h}_v) \cdot \cos(\mathbf{h}_v, \mathbf{h}_w) - \sqrt{(1 - \cos(\mathbf{h}_u, \mathbf{h}_v)^2) \cdot (1 - \cos(\mathbf{h}_v, \mathbf{h}_w)^2)}$$

In such cases, a similarity measure becomes inherently transitive, i.e., if $\mathbf{h}_u \sim \mathbf{h}_v$ and $\mathbf{h}_v \sim \mathbf{h}_w$, then $\mathbf{h}_u \sim \mathbf{h}_w$ also holds ($\sim$ for similar). For instance, $\cos(\mathbf{h}_u, \mathbf{h}_v)$ is close to 1 means $\mathbf{h}_u \sim \mathbf{h}_v$, and $\cos(\mathbf{h}_u, \mathbf{h}_v)$ is close to $-1$ means $\mathbf{h}_u$ and $\mathbf{h}_v$' are highly dissimilar. Given these properties, by triangle inequality above, if both $\cos(\mathbf{h}_u, \mathbf{h}_v)$ and $\cos(\mathbf{h}_v, \mathbf{h}_w)$ are close to 1, then so is $\cos(\mathbf{h}_u, \mathbf{h}_w)$.

However, the link prediction task is generally not transitive. Specifically, if $u \sim v$ and $v \sim w$, it does not necessarily imply that $u \sim w$. In Table 1, we present the transitivity ratio for various benchmark datasets used in link prediction. We define the *transitivity ratio* as the proportion of node pairs $(u, w)$ that have an edge between them out of all node pairs that share at least one common neighbor $v$. The data shows that the link prediction task is highly non-transitive, as most node pairs with a common neighbor do not form a link. Consequently, using any distance or similarity-based operation as a prediction head can significantly hinder the performance of GNNs.

This is why using individual node representations to represent node pairs is not optimal as they can bring restrictions for GNNs. Instead, learning representations of node pairs (or edge representations) can be more effective for link prediction tasks. From this perspective, our approach, PROXI, can be seen as direct node pair embedding where the coordinates are individual proximity indices.

For further discussion on the effect of similarity-based decoders, we refer to recent studies (Zhu et al., 2024; Cho, 2024). Furthermore, in Wang et al. (2022a), the authors evaluated the performance of various decoders for GNNs and demonstrated that models using similarity-based prediction heads (e.g., dot product) exhibited suboptimal performance. Their comparison table can be found in Table 12 in our appendix.

Table 1: Transitivity ratios of the datasets

| TEXAS | WISC. | CHAM. | SQUIR. | CROC. | CORA | CITESEER | PHOTO |
|-------|-------|-------|--------|-------|------|----------|-------|
| 0.046 | 0.049 | 0.009 | 0.002  | 0.002 | 0.092 | 0.145   | 0.010 |

## 4 Experiments

### 4.1 Experimental Setup

**Datasets.** In our experiments, we used twelve benchmark datasets for link prediction tasks. All the datasets are used in the transductive setting like most other baselines in the domain. The dataset statistics are given in Table 2. The details of the datasets are given in Appendix A.

**Experiment Settings.** To compare our model's performance, we adopted the common method proposed in Lichtenwalter et al. (2010) with 85/5/10 split for all datasets except OGB datasets which come with their own predefined training and test sets. To expand the comparison baselines, we also report the performance of our model with different split 70/10/20 in Table 9 in the Appendix.

**Proximity Indices.** In all datasets except OGBL-COLLAB, we used the same proximity indices described in Section 3.2. Since OGBL-COLLAB is dynamic and weighted, we needed to adjust our domain proximity indices to adapt our method to this context. The total number of indices/parameters used for each dataset is given in Table 11. We gave the details of these proximity indices for each dataset in Appendix C.

**Metrics.** In all datasets, we used the common performance metric, AUC. In addition, in OGBL datasets and second split 70/10/20, to compare with the recent baselines, we further used Hits@K metric which counts the ratio of positive links that are ranked at K-place or above, after ranking each true link among a set of 100,000 randomly sampled negative links (Hu et al., 2021b).

**Hyperparameter Settings.** In our study, XGBoost acts as the primary machine learning tool. The optimization objective is defined as rank: pairwise, with logloss operating as the evaluation metric. When assessing outcomes through the AUC metric, we configure the maximum tree depth of 5, the learning rate varying in [0.01, 0.05], the number of estimators at 1000, and the regularization parameter lambda set to 10.0. For the more demanding metric, Hits@20, modifications are implemented. Notably, the maximum tree depth is upgraded to 5, the colsample bytree ratio is adjusted to 1, the learning rate is set to 0.1, and lambda is set to 1.0, while other parameters remain consistent with those utilized for the AUC metric. Similarly, for the Hits@50 metric, the maximum tree depth is reset to 11, the learning rate is increased to 0.5, and lambda is set to 1.0, with all other hyperparameters aligned with those used for the AUC metric. Lastly, in the context of the Hits@100 metric, adaptions are made within the AUC hyperparameter setting. These adjustments encompass changing the maximum tree depth to 5, setting the learning rate to 0.3, adjusting the subsample ratio to 0.5, and the colsample bytree ratio to 1.0, while lambda is maintained at 1.0.

Table 2: Characteristics of our benchmark datasets for link prediction. FV Type represents the type of the node attribute vector provided.

| Datasets | Nodes | Edges | Classes | Attributes | FV Type | N. Hom. | E. Hom. |
|----------|-------|-------|---------|-----------|---------|---------|---------|
| CORA | 2,708 | 5,429 | 7 | 1,433 | Binary | 0.83 | 0.81 |
| CITESEER | 3,312 | 4,732 | 6 | 3,703 | Binary | 0.71 | 0.74 |
| PUBMED | 19,717 | 44,338 | 3 | 500 | Binary | 0.79 | 0.80 |
| PHOTO | 7,650 | 119,081 | 8 | 745 | Binary | 0.77 | 0.83 |
| COMPUTERS | 13,752 | 245,861 | 10 | 767 | Binary | 0.82 | 0.78 |
| OGBL-COLLAB | 235,868 | 1,285,465 | – | 128 | Real | – | – |
| OGBL-PPA | 576,289 | 30,326,273 | 58 | – | – | – | – |
| TEXAS | 183 | 295 | 5 | 1,703 | Binary | 0.09 | 0.41 |
| CORNELL | 183 | 280 | 5 | 1,703 | Binary | 0.39 | 0.57 |
| WISCONSIN | 251 | 466 | 5 | 1,703 | Binary | 0.15 | 0.45 |
| CHAMELEON | 2,277 | 31,421 | 5 | 2,325 | Binary | 0.25 | 0.28 |
| SQUIRREL | 5,201 | 198,493 | 5 | 2,089 | Binary | 0.22 | 0.24 |
| CROCODILE | 11,631 | 170,918 | 5 | 500 | Binary | 0.25 | 0.25 |

Table 3: **Link Prediction Results.** AUC results on benchmark datasets with baseline performances reported by Zhou et al. (2022). Additional performance comparisons are given in Table 9 in the Appendix.

| Models | TEXAS | WISC. | CHAM. | SQUIR. | CROC. | CORA | CITESEER | PHOTO |
|---|---|---|---|---|---|---|---|---|
| *Hom.* | 0.39 | 0.15 | 0.10 | 0.25 | 0.22 | 0.83 | 0.72 | 0.77 |
| CN (Newman, 2001) | 53.0±6.2 | 61.3±0.9 | 95.3±0.6 | 96.8±0.4 | 89.8±1.2 | 75.7±1.3 | 69.7±2.3 | 97.1±0.4 |
| AA (Adamic & Adar, 2003) | 53.1±6.2 | 61.5±0.8 | 95.8±0.6 | 97.1±0.4 | 90.6±1.1 | 75.9±1.5 | 69.8±2.2 | 97.4±0.4 |
| VGAE (Alemi et al., 2016) | 68.6±4.2 | 71.3±4.6 | 98.5±0.1 | 98.2±0.1 | 98.8±0.1 | 96.6±0.2 | 97.3±0.2 | 94.9±0.8 |
| SEAL (Zhang & Chen, 2018) | 73.9±1.6 | 72.3±2.7 | **99.5±0.1** | - | - | 90.4±1.1 | 97.0±0.5 | - |
| ARGVA (Pan et al., 2018) | 67.4±6.1 | 67.6±3.0 | 96.5±0.2 | 93.6±0.3 | 94.1±0.5 | 92.7±1.3 | 94.8±0.4 | 89.7±2.3 |
| DisenGCN (Ma et al., 2019) | 72.1±4.8 | 75.1±3.4 | 97.7±0.1 | 94.5±0.2 | 96.4±0.4 | 96.6±0.3 | 96.8±0.2 | 95.2±1.3 |
| FactorGNN (Yang et al., 2020) | 58.7±4.1 | 68.8±9.0 | 98.3±0.3 | 96.9±0.4 | 97.6±0.4 | 92.3±1.4 | 87.8±3.6 | 97.2±0.1 |
| GPR-GNN (Chien et al., 2020) | 76.3±2.5 | 80.1±4.5 | 98.7±0.1 | 96.0±0.3 | 96.7±0.1 | 94.8±0.3 | 96.0±0.3 | 97.0±0.2 |
| FAGCN (Bo et al., 2021) | 68.7±7.3 | 73.7±4.9 | 93.8±2.6 | 94.8±0.3 | 95.3±0.2 | 91.8±3.4 | 89.2±5.6 | 94.0±1.9 |
| LINKX (Lim et al., 2021) | 75.8±4.7 | 80.1±3.8 | 98.8±0.1 | 98.1±0.3 | 99.0±0.1 | 93.4±0.3 | 93.5±0.5 | 97.0±0.2 |
| VGNAE (Ahn & Kim, 2021) | 78.9±3.0 | 70.3±1.2 | 95.4±0.2 | 89.4±0.1 | 90.5±0.4 | 89.2±0.7 | 95.5±0.6 | 79.5±0.2 |
| DisenLink (Zhou et al., 2022) | 80.7±4.0 | 84.4±1.9 | 99.4±0.0 | 98.3±0.1 | 99.1±0.1 | **97.1±0.4** | **98.3±0.3** | 97.9±0.1 |
| VDGAE (Fu et al., 2023) | 81.3±8.5 | 85.0±4.8 | 96.8±0.1 | 95.8±0.1 | 93.9±0.2 | 95.9±0.4 | 97.8±0.3 | 94.7±0.1 |
| **PROXI** | **84.6±2.4** | **85.8±2.2** | 98.8±0.1 | **98.3±0.0** | **99.6±0.0** | 95.8±0.6 | 95.7±0.6 | **99.1±0.1** |

**Implementation and Runtime.** We ran experiments on a single machine with 12th Generation Intel Core i7-1270P vPro Processor (E-cores up to 3.50 GHz, P-cores up to 4.80 GHz), and 32Gb of RAM (LPDDR5-6400MHz). End-to-end runtime (computing proximity indices and ML classifier) for OGBL-COLLAB is 30 minutes. Similarly, the runtime is 15 minutes for COMPUTERS (250K nodes) for all indices. The computational complexities of our similarity indices are $\mathcal{O}(|\mathcal{V}|.k^3)$ where $|\mathcal{V}|$ is the total number of nodes and $k$ is the maximum degree in the network (Martínez et al., 2016). By utilizing the powers of adjacency matrix, we would significantly reduced the computational time. We would like to thank the reviewer for this insightful suggestion. We provide our code at `https://github.com/workrep20232/PROXI`

## 4.2 Results

**Baselines.** We compare the link prediction performance of our model PROXI against the common embedding methods and GNNs. The baseline performances are reported from the references indicated in the captions of Tables 3, 4 and 9. For OGB-COLLAB, we used the baseline performances from Li et al. (2023) and further reported the performance of the current leader (Wang et al., 2022b) at OGB Leaderboard (Hu et al., 2021b).

**Results.** We give our results in the Tables 3, 4 and 9. We have five heterophilic and three homophilic datasets. In Table 3, we observe that our PROXI model outperforms SOTA GNN models in four out of five

Table 4: Performances for OGBL datasets with baselines by (Li et al., 2023).

| | OGBL-COLLAB | | OGBL-PPA | |
|---|---|---|---|---|
| Models | Hits@50 | AUC | Hits@100 | AUC |
| MF (Menon & Elkan, 2011) | 41.81±1.67 | 83.75±1.77 | 28.40±4.62 | 99.46±0.10 |
| N2Vec (Grover & Leskovec, 2016) | 49.06±1.04 | 96.24±0.15 | 26.24±0.96 | 99.77±0.00 |
| GCN (Kipf & Welling, 2016a) | 54.96±3.18 | 97.89±0.06 | 29.57±2.90 | 99.84±0.03 |
| GAT (Veličković et al., 2018) | 55.00±3.28 | 97.11±0.09 | OOM | OOM |
| SAGE (Hamilton et al., 2017) | 59.44±1.37 | 98.08±0.03 | 41.02±1.94 | 99.82±0.00 |
| SEAL (Zhang & Chen, 2018) | 63.37±0.69 | 95.65±0.29 | 48.40±5.61 | 99.79±0.02 |
| Neo-GNN (Yun et al., 2021) | 66.13±0.61 | **98.23±0.05** | 48.45±1.01 | 97.30±0.14 |
| BUDDY (Chamberlain et al., 2022) | 64.59±0.46 | 96.52±0.40 | 47.33±1.96 | 99.56±0.02 |
| NCN (Wang et al., 2023) | 63.86±0.51 | 97.83±0.04 | 62.63±1.15 | 99.95±0.01 |
| NCNC (Wang et al., 2023) | 65.97±1.03 | 98.20±0.05 | 62.61±0.76 | **99.97±0.01** |
| OGB Leader (Hu et al., 2021b) | 71.29±0.18 | – | **65.24±1.50** | – |
| **PROXI** | **76.50±0.27** | 97.24±0.03 | 50.36±0.76 | 99.90±0.00 |

heterophilic datasets. Similarly, in homophilic setting, while outperforming SOTA in one dataset, it gives very competitive results in the other two.

In OGBL datasets (Table 4), PROXI outperforms all SOTA GNNs and OGBL leader in the OGBL-COLLAB dataset for the OGB metric Hits@50. In OGBL-PPA, the AUC performance of PROXI stands shoulder to shoulder with the best-performing models.

Finally, in our second split for comparison with additional baselines (Table 9), PROXI again outperforms SOTA GNNs in two out of three homophilic datasets, while giving highly competitive performance in the third one.

Table 5: AUC results comparing GNN models w/out PROXI integration on the CORA and TEXAS datasets.

| Dataset | Backbone | GNN | PROXI-GNN | Up+ |
|---------|----------|-----|-----------|-----|
| **CORA** | GCN | $87.71\pm2.06$ | $92.90\pm0.92$ | **5.19** |
| | GAT | $86.86\pm1.78$ | $94.56\pm0.87$ | **7.70** |
| | SAGE | $87.67\pm1.95$ | $95.11\pm1.32$ | **7.44** |
| | LINKX | $93.40\pm0.30$ | $93.57\pm0.76$ | **0.17** |
| **TEXAS** | GCN | $63.34\pm5.63$ | $75.35\pm4.97$ | **11.01** |
| | GAT | $65.65\pm3.75$ | $76.84\pm4.52$ | **11.19** |
| | SAGE | $69.58\pm7.14$ | $77.10\pm4.26$ | **7.52** |
| | LINKX | $75.80\pm4.70$ | $79.59\pm5.53$ | **3.79** |

**PROXI-GNNs.**   While our results show that GNNs are not significantly better than traditional models, a natural question is how to use PROXI indices with current ML models. As discussed before, the link prediction task is a binary problem in the space of node pairs, and providing information on node pairs is very crucial to getting effective prediction heads. We took this path, and concatenate our proximity indices $\{\alpha(u,v)\}$ through MLP to prediction heads $\mathbf{h}_u \odot \mathbf{h}_v$ (Hadamard product) of classical GNN models (See Appendix B for details). Our results are outstanding as we see consistent major improvements for both homophilic and heterophilic settings (Table 5).

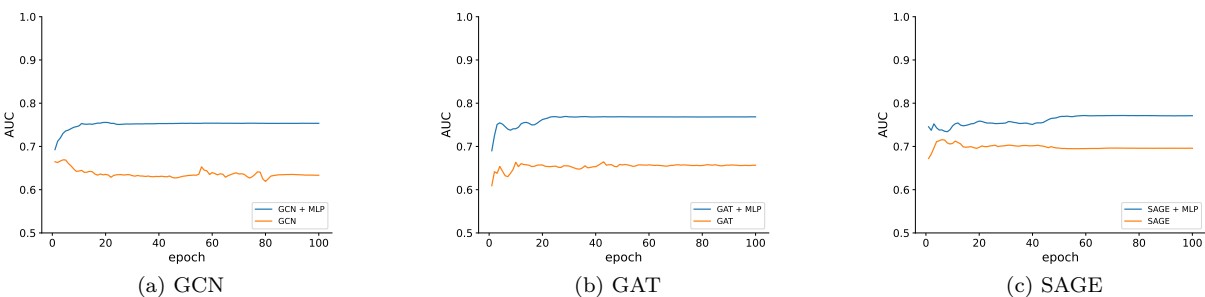

(a) GCN        (b) GAT        (c) SAGE

Figure 1: **PROXI-GNN.** Performance comparison of vanilla-GNNs (orange) and PROXI-GNNs (blue) on TEXAS dataset.

**Ablation Study.**   We made two ablation studies on our method. In the first one, we evaluated the performances of our structural and domain indices alone (Table 6). In both homophilic and heterophilic settings, the outcomes are varied: we observe that one set of indices is not better than the other in general. Nonetheless, a notable consistent finding across all datasets is the synergistic effect of these indices. *Their combination consistently improves overall performance.*

In the second ablation study, we evaluate the performance of various ML classifiers using PROXI indices (Table 8). The comparable results across all classifiers demonstrate that our PROXI indices are both highly effective and model-agnostic.

**Index Importance Scores.**   In Table 7, we present the importance of individual proximity indices in our model across various datasets. We obtain index importance scores through XGBoost's built-in function: feature importance. Our index set proves to be highly versatile, effectively adapting to the unique characteristics of each dataset. Notably, certain indices exhibit substantial importance in specific datasets, while their

Table 6: **Structural vs. Domain Indices.** AUC results for our model for different proximity index subsets.

| Indices | CORA | CITESEER | PUBMED | PHOTO | COMP. | TEXAS | WISC. | CHAM. | SQUIR. | CROC. |
|---------|------|----------|--------|-------|-------|-------|-------|-------|--------|-------|
| Structural only | $88.66\pm1.10$ | $81.33\pm1.21$ | $92.60\pm0.34$ | $93.46\pm0.15$ | $98.60\pm0.03$ | $68.51\pm3.35$ | $72.92\pm3.35$ | $97.46\pm0.19$ | $97.86\pm0.05$ | $99.00\pm0.03$ |
| Domain only | $91.10\pm0.94$ | $92.77\pm0.89$ | $87.40\pm0.34$ | $92.98\pm0.16$ | $88.97\pm0.14$ | $83.09\pm2.23$ | $81.93\pm2.28$ | $90.44\pm0.47$ | $82.97\pm0.13$ | $92.73\pm0.09$ |
| All Indices | $\mathbf{95.83\pm0.59}$ | $\mathbf{95.71\pm0.58}$ | $\mathbf{96.08\pm0.16}$ | $\mathbf{99.15\pm0.05}$ | $\mathbf{98.93\pm0.03}$ | $\mathbf{84.61\pm2.37}$ | $\mathbf{85.84\pm2.18}$ | $\mathbf{98.77\pm0.10}$ | $\mathbf{98.27\pm0.04}$ | $\mathbf{99.56\pm0.02}$ |

Table 7: **Index Importance.** For each dataset, the importance weights of our indices for XGBoost. The top three important indices in each dataset are given in **blue**, **green**, and **red** respectively. Newly introduced indices are marked in blue.

|  | CORA | CITES. | PUBMED | PHOTO | COMP. | TEXAS | WISC. | CHAM. | SQUIR. | CROC. |
|---|---|---|---|---|---|---|---|---|---|---|
| graph distance | 0.0385 | 0.0217 | **0.5976** | 0.0328 | **0.0275** | 0.0546 | 0.0709 | 0.1199 | **0.3835** | 0.0953 |
| # 2-paths | **0.0581** | 0.0607 | 0.0082 | 0.0066 | 0.0128 | 0.0165 | 0.0300 | 0.0159 | 0.0123 | 0.0089 |
| Adamic Adar | 0.0435 | 0.0426 | 0.0087 | **0.7436** | **0.7762** | **0.0798** | **0.1492** | 0.0344 | **0.4040** | 0.0282 |
| Jaccard | 0.0050 | 0.0049 | 0.0016 | 0.0063 | 0.0066 | 0.0531 | 0.0411 | 0.0069 | 0.0098 | 0.0055 |
| Salton | 0.0036 | 0.0035 | 0.0017 | 0.0028 | 0.0023 | 0.0445 | 0.0283 | 0.0090 | 0.0041 | 0.0024 |
| Sorensen | 0 | 0 | 0 | 0 | 0 | 0.0423 | 0.0332 | 0.0126 | 0.0182 | 0.0114 |
| # 3-paths | **0.1179** | **0.3056** | **0.1624** | 0.0117 | 0.0210 | 0.0479 | **0.1047** | **0.4093** | **0.0525** | **0.2424** |
| 3-Jaccard | 0.0120 | 0.0406 | 0.0431 | 0.0039 | 0.0048 | 0.0355 | 0.0224 | 0.0123 | 0.0215 | 0.0076 |
| 3-Salton | 0.0113 | 0.0654 | **0.0604** | **0.0455** | 0.0272 | 0.0467 | 0.0349 | **0.1402** | 0.0178 | **0.3649** |
| 3-Sorensen | 0.0051 | 0.0078 | 0.0054 | 0.0037 | 0.0047 | 0.0507 | 0.0302 | 0.0092 | 0.0103 | 0.0083 |
| common digits | 0.0043 | 0.0052 | 0.0067 | 0.0040 | 0.0039 | 0.0496 | 0.0729 | **0.1378** | 0.0247 | **0.1058** |
| common class | **0.6254** | **0.1304** | 0.0386 | 0.0417 | 0.0266 | **0.1859** | 0.0652 | 0.0057 | 0.0018 | 0.0083 |
| norm. digits | 0.0264 | **0.2517** | 0.0356 | 0.0025 | 0.0027 | 0.0490 | 0.0497 | 0.0330 | 0.0112 | 0.0320 |
| class identifier | 0.0489 | 0.0600 | 0.0300 | **0.0949** | **0.0837** | **0.2440** | **0.2672** | 0.0539 | 0.0283 | 0.0791 |

impact is minimal in others. Adamic Adar emerges as the most important index in four out of ten datasets. Notably, in the remaining six datasets, our newly introduced proximity indices take precedence in five of them. Moreover, the table underscores the significant variation in index importance, with no particular subset of indices dominating others. *Overall, the synergy between domain and structural proximity indices emerges as a pivotal factor contributing to improved performance.*

Table 8: AUC results of PROXI for different ML Classifiers.

| ML Classifier | CORA | CITESEER | PUBMED | TEXAS | WISC. | CHAM. |
|---|---|---|---|---|---|---|
| Logistic Reg. | $94.89_{\pm0.57}$ | $95.30_{\pm0.82}$ | $95.68_{\pm0.14}$ | $81.86_{\pm3.80}$ | $82.37_{\pm3.74}$ | $97.83_{\pm0.18}$ |
| Naive Bayes | $94.88_{\pm0.49}$ | $92.55_{\pm0.58}$ | $94.77_{\pm0.16}$ | $76.84_{\pm3.56}$ | $74.04_{\pm1.95}$ | $96.64_{\pm0.20}$ |
| QDA | $95.23_{\pm0.53}$ | $94.06_{\pm0.58}$ | $94.70_{\pm0.23}$ | $81.16_{\pm3.07}$ | $79.29_{\pm1.45}$ | $96.71_{\pm0.20}$ |
| XGBoost | $\mathbf{95.83_{\pm0.59}}$ | $\mathbf{95.71_{\pm0.58}}$ | $\mathbf{96.08_{\pm0.16}}$ | $\mathbf{84.61_{\pm2.37}}$ | $\mathbf{85.84_{\pm2.18}}$ | $\mathbf{98.77_{\pm0.10}}$ |

**Discussion.** Our experiments show that our simple ML model, which is a combination of proximity indices with a tree-based ML classifier, outperforms or gives an on-par performance with most of the current GNN models in benchmark datasets. We would like to note that another traditional method (Adamic-Adar & Edge Proposal Set) by Singh et al. (2021) has a very high ranking in the OGBL-COLLAB leaderboard. These findings may be seen as unexpected since GNN models are generally perceived as the frontrunners in graph representation learning. However, as recent theoretical studies suggest, our results underscore the pressing need for fresh and innovative approaches within the realm of GNNs to address unique challenges in graph representation learning and improve performance.

**Limitations.** The main limitation in our approach comes from the domain proximity indices depending on the format of the node attribute vectors and the context. Unfortunately, there is no general rule in this part, as the context of the dataset plays a crucial part in extracting useful domain indices. However, considering node attribute vectors as node embedding in the attribute space, we plan to use GNNs to obtain the most effective domain proximity indices by formulating the question in terms of learnable parameters. In our future projects, we aim to further explore this direction.

## 5 Conclusion

In this paper, motivated by the theoretical studies, we have compared the performance of state-of-the-art GNN models with conventional methods in link prediction tasks. Our computationally efficient model, which utilizes a collection of structural and domain proximity indices, outperforms or gives highly competitive results

with state-of-the-art GNNs across both small/large benchmark datasets as well as homophilic/heterophilic settings. Furthermore, when combined, our PROXI indices significantly enhances the performances of classical GNN methods. Our results not only empirically verify recent theoretical insights but also suggest significant untapped potential within GNNs. Looking ahead, we aim to delve into building GNN models tailored for link prediction, which utilizes edge embeddings. We intend to integrate our proximity indices to obtain stronger edge representations, thus enhancing the overall performance significantly.

### Acknowledgements

This work was partially supported by National Science Foundation under grants DMS-2220613 and DMS-2229417. The authors acknowledge the Texas Advanced Computing Center (TACC) at UT Austin for providing computational resources that have contributed to the research results reported within this paper. `http://www.tacc.utexas.edu`.

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

## Appendix

In this part, we provide extra experimental results and further details on our method. In Appendix A, we provide dataset details, the performance of our model with different ML classifiers (Table 8), and further performance results (Table 9). In Appendix B, we provide the details of how PROXI indices can boost the performance of GNN models. In Appendix C, we provide details of our proximity indices used for all datasets. Finally, in Appendix D.2, we discuss the relation between transitivity and heterophily.

## A    Datasets

In our experiments, we used twelve benchmark datasets for link prediction tasks. All the datasets are used in the transductive setting like most other baselines in the domain. The dataset statistics are given in Table 2.

The citation network datasets, namely CORA, CITESEER, and PUBMED are introduced in (Yang et al., 2016), and they serve as valuable benchmark datasets for research in the field of semi-supervised learning with graph representation learning. Within these datasets, individual nodes correspond to distinct documents, while the edges between them symbolize citation links, elucidating the interconnectedness of scholarly works within these domains.

In the context of co-purchasing networks, the benchmark datasets, PHOTO, and COMPUTERS, are introduced in (Shchur et al., 2018) representing the sales network at Amazon. In these networks, nodes correspond to various products, while edges signify the frequent co-purchasing of two products. The primary objective of this study is to leverage product reviews, represented as bag-of-words node attributes, to establish a mapping between individual goods and their respective product categories, thus addressing a fundamental categorization task within the context of these interconnected networks.

Next, the OGBL-COLLAB dataset is a part of the library of large benchmark datasets, namely Open Graph Benchmark (OGB) collection (Hu et al., 2020; 2021a). This is an undirected graph, representing a subset of the collaboration network between authors indexed by Microsoft Academic Graph (MAG) (Wang et al., 2020). Each node represents an author and edges indicate the collaboration between authors. All nodes come with 128-dimensional attributes, obtained by averaging the word embeddings of papers that are published by the authors. All edges are associated with two meta-information: the year and the edge weight, representing the number of co-authored papers published in that year. The graph can be viewed as a dynamic multi-graph since there can be multiple edges between two nodes if they collaborate in more than one year.

Another OGB dataset is OGBL-PPA, which represents an unweighted, undirected graph structure. Nodes in this complex network carefully depict proteins from a wide range of 58 different species, demonstrating the dataset's extensive coverage of the biological environment. Furthermore, this complex network's edges encode a wide range of biologically significant relationships between proteins. These relationships cover a wide range of interactions, such as physical contacts, co-expression patterns, homology determinations, and the defining of genomic regions (Szklarczyk et al., 2019).

Table 9: AUC and Hits@20 results on homophilic datasets with baselines by Zhao et al. (2022) for 70:10:20 split.

| Models | CORA | | CITESEER | | PUBMED | |
|---|---|---|---|---|---|---|
| | AUC | Hits@20 | AUC | Hits@20 | AUC | Hits@20 |
| Node2Vec (Grover et al., 2016) | 84.49±0.49 | 49.96±2.51 | 80.00±0.68 | 47.78±1.72 | 80.32±0.29 | 39.19±1.02 |
| VGAE (Kipf & Welling, 2016b) | 88.68±0.40 | 45.91±3.38 | 85.35±0.60 | 44.04±4.86 | 95.80±0.13 | 23.73±1.61 |
| GCN (Kipf & Welling, 2017) | 90.25±0.53 | 49.06±1.72 | 71.47±1.40 | 55.56±1.32 | 96.33±0.80 | 21.84±3.87 |
| GSAGE (Hamilton et al., 2017) | 90.24±0.34 | 53.54±2.96 | 87.38±1.39 | 53.67±2.94 | 96.78±0.11 | 39.13±4.41 |
| SEAL (Zhang & Chen, 2018) | 92.55±0.50 | 51.35±2.26 | 85.82±0.44 | 40.90±3.68 | 96.36±0.28 | 28.45±3.81 |
| JKNet (Xu et al., 2018) | 89.05±0.67 | 48.21±3.86 | 88.58±1.78 | 55.60±2.17 | 96.58±0.23 | 25.64±4.11 |
| MVGRL (Hassani et al., 2020) | 75.07±3.63 | 19.53±2.64 | 61.20±0.55 | 14.07±0.79 | 80.78±1.28 | 14.19±0.85 |
| LGLP (Cai et al., 2021) | 91.30±0.05 | 62.98±0.56 | 89.41±0.13 | 57.43±3.71 | – | – |
| CFLP (Zhao et al., 2022) | 93.05±0.24 | **65.57±1.05** | 92.12±0.47 | 68.09±1.49 | **97.53±0.17** | **44.90±2.00** |
| PROXI | **94.70±0.63** | 61.24±2.90 | **94.97±0.29** | **71.94±1.82** | 94.66±0.13 | 42.10±2.21 |

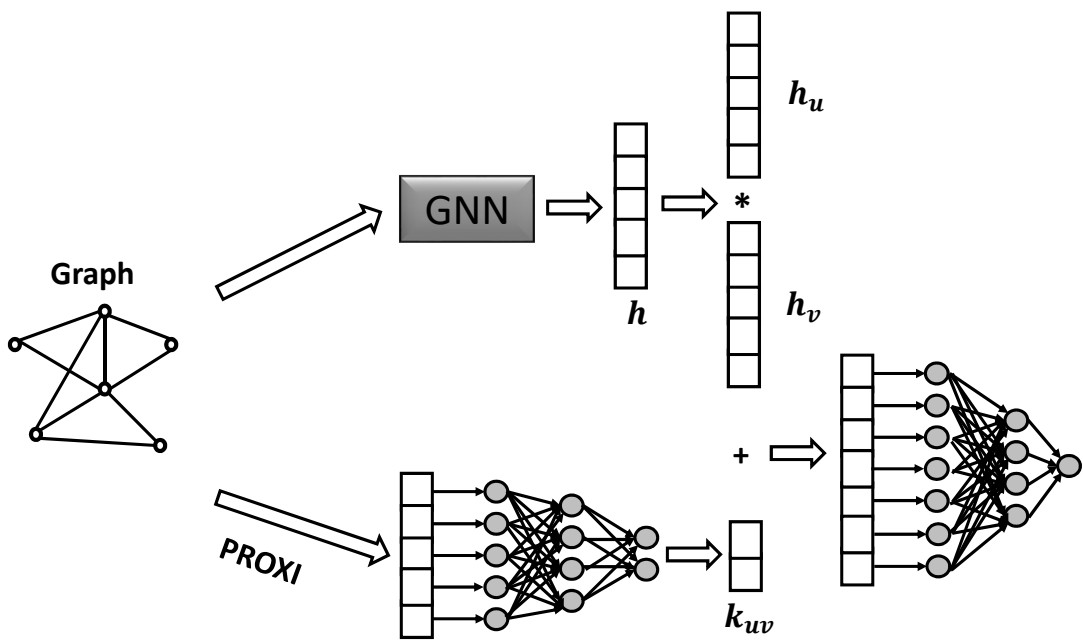

Figure 2: **PROXI-GNN:** The flowchart of the model we used to integrate PROXI indices with GNNs. In the image '$*$' represents the Hadamard product $\mathbf{h}_u \odot \mathbf{h}_v$ and '$+$' represents concatenation.

The datasets WISCONSIN and TEXAS (Pei et al., 2020) are heterophilic website networks that are acquired via CMU. Web pages are represented as nodes in these networks, and the hyperlinks that connect them as edges. The bag-of-words representation of the corresponding web pages defines the node attributes.

Finally, the Wikipedia heterophilic webpage structures that are represented by the networks CROCODILE, CHAMELEON, and SQUIRREL (Pei et al., 2020) are each focused on a particular topic, as denoted by the name of the corresponding dataset. Nodes in these networks are individual websites, while edges are hyperlinks that connect them. The attributes assigned to every node consist of a variety of educational nouns taken from Wikipedia articles.

## B Integrating PROXI indices with GNNs

In this section, we study the integration of PROXI indices to enhance the performance of GCN, GAT, SAGE and LINKX models, and give details of our results presented in Table 5. The conventional approach without incorporating the indices follows the standard procedure. Initially, employing one of the GNN models mentioned, node embeddings denoted as $\mathbf{h}_v$, are generated over the training set using the node attributes available in the dataset. Next, within the predictor, for each edge $(u, v)$, the element-wise multiplication (Hadamard product) $\mathbf{h}_u \odot \mathbf{h}_v = (h_u^1 \cdot h_v^1, h_u^2 \cdot h_v^2, ..., h_u^N \cdot h_v^N)$ is computed, where $h_u^i$ and $h_v^i$ represent the $i^{th}$ element of node embeddings $\mathbf{h}_u$ and $\mathbf{h}_v$ respectively. This resultant edge representation is then passed through a 3-layer MLP to obtain the final edge prediction. The flowchart of our simple model integrating PROXI indices with GNNs is presented in Figure 2.

To test the effect of PROXI indices in the performance GNN, we introduce a PROXI enhanced GNN model, *PROXI-GNN*. Our main motivation as discussed earlier, learning node representations is a meaningful task for node classification task, however, it is not very suitable for link prediction task. Instead, the goal is to learn representations of node pairs, and consider this as a binary classification problem (positive vs. negative edge). Our proximity indices directly provides embeddings of node pairs. We will use this direct information in GNNs by enhancing their prediction heads for node pairs. Specifically, within the predictor, our model takes the indices of a node pair $(u, v)$ and processes them through a 3-layer MLP to extract meaningful representations.

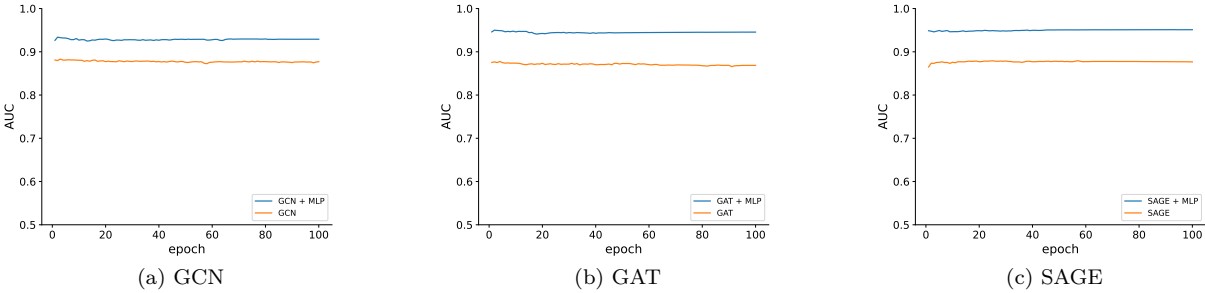

Figure 3: Performance comparison of vanilla-GNNs (orange) and PROXI-GNNs (blue) on CORA dataset.

These MLP layers help in capturing intricate patterns and characteristics present in the indices. Following this, the resulting edge embedding from the MLP layers is concatenated with the element-wise product $\mathbf{h}_u \odot \mathbf{h}_v$ obtained earlier, and then it is passed through 3-layer MLP as above. This concatenation operation combines the learned indices with the node embeddings, creating a fused representation that captures both node-level and edge-level information. Our methodology seeks to utilize the extra information included in the relationships between nodes, as encoded in the indices, by integrating the indices in this way. With both node and edge information included, this enhanced representation offers a more complete picture of the network structure, which could enhance prediction performance. Our model's ability to accurately forecast the existence or attributes of edges is improved by incorporating indices directly into the prediction process. This helps the model to better capture the intricate interconnections and dependencies inside the graph.

In Figure 3, we present a comparative analysis between the standard algorithm, GNN, and our proposed modified algorithm, PROXI-GNN, on the Cora dataset, employing the before mentioned GNN models. The visualization illustrates the performance comparison across different GNN models, highlighting the impact of integrating our edge feature vectors into the models.

From the depicted results, it is evident that incorporating our edge feature vectors yields notable improvements in performance across all GNN models examined. Specifically, our modified algorithm PROXI-GNN consistently outperforms the standard algorithm by a margin of at least 5% in terms of predictive accuracy. This substantial enhancement underscores the significance of integrating edge feature information into the modeling process. Same improvement can be noticed for heterophilic dataset Texas in Figure 1, where the AUC is considerably improved by 8%.

The observed improvements in performance serve as compelling evidence of the efficacy of our proposed approach. By leveraging the additional contextual information embedded in the indices, our modified algorithm effectively enhances the models' ability to capture intricate relationships and dependencies within the graph structure. This enhancement ultimately leads to more accurate and robust predictions, as demonstrated by the substantial performance gains across the GNN models evaluated.

Overall, the outcomes shown in Figures 3 and 1 highlight the significance and effectiveness of the modifications we suggested. The noteworthy enhancements in performance that may be obtained by using our proposed edge feature vectors underscore the possibility of our methodology to augment the predictive powers of GNN models, especially in situations where edge-level data is pivotal to the underlying graph structure.

In Table 5, we present a performance comparison of GNN models with PROXI-GNN on two distinct datasets, CORA (homophilic) and TEXAS (heterophilic). Notably, the inclusion of our indices demonstrates significant improvements in classification AUC, i.e., **up to 8% in CORA, and 11% in TEXAS.** This substantial increase in AUC suggests that the incorporation of our indices enhances the model's ability to learn from graph structures. Similar results are observed on the heterophilic dataset TEXAS. These results underscore the importance of considering proximity indices in graph-based learning tasks, showcasing the efficacy of our PROXI method in improving GNN performance.

### B.1 GNN Models with Proximity Indices

Several recent approaches, including BUDDY/ELPH (Chamberlain et al., 2022), the Labeling Trick (Zhang et al., 2021) and HL-GNN (Zhang et al., 2024), have addressed the limitations of GNNs in link prediction tasks by focusing on generating meaningful representations for node sets or node pairs. BUDDY/ELPH enhances MPNNs by incorporating subgraph information into the message-passing process, implicitly capturing many structural indices akin to those used in our model. However, it does not utilize the domain-specific indices central to our methodology, which significantly improve performance. These indices provide a direct and dataset-adaptive mechanism to inform the ML model about node-pair relationships, resulting in superior performance on OGBL-COLLAB and OGBL-PPA datasets (Table 10). In contrast to BUDDY's node-centric approach, we are developing a novel GNN model for link prediction that learns edge (or node-pair) representations, using our indices as initial messages, thereby offering a complementary perspective to the challenges of link prediction.

The Labeling Trick similarly focuses on addressing the challenges of generating meaningful node-set representations by leveraging node labeling schemes, such as the zero-one labeling. While this aligns with some of our domain-specific indices, including the "class identifier" and "common class" indices, it does not natively incorporate others, such as "common digits" and "normalized digits," without additional adaptations. Notably, our PROXI model outperforms all GNN models enhanced with the Labeling Trick, as reported in Zhang et al. (2021), on both OGBL-PPA and OGBL-COLLAB datasets

Table 10: Comparison of model performance on OGBL-PPA (Hits@50) and OGBL-COLLAB (Hits@100) datasets.

| Model | OGBL-PPA | OGBL-COLLAB |
|---|---|---|
| BUDDY (Chamberlain et al., 2022) | $47.33_{\pm 1.96}$ | $64.59_{\pm 0.46}$ |
| GCN+DE (Zhang et al., 2021) | $36.48_{\pm 3.78}$ | $64.44_{\pm 0.29}$ |
| GCN+DRNL (Zhang et al., 2021) | $45.24_{\pm 3.95}$ | $64.40_{\pm 0.45}$ |
| SEAL (Zhang et al., 2021) | $48.80_{\pm 3.16}$ | $64.74_{\pm 0.43}$ |
| HL-GNN (Zhang et al., 2024) | $\mathbf{56.77_{\pm 0.84}}$ | $68.11_{\pm 0.54}$ |
| PROXI | $50.36_{\pm 0.76}$ | $\mathbf{76.50_{\pm 0.27}}$ |

(Table 10). This performance advantage highlights the effectiveness of directly integrating domain-specific indices into the prediction mechanism. The comparisons underscore the flexibility and robustness of our approach, which addresses fundamental challenges in link prediction more comprehensively than existing methods.

Similarly, HL-GNN (Zhang et al., 2024) effectively incorporates several structural indices that overlap with our methodology into its message aggregation scheme, as outlined in (Zhang et al., 2024, Table 2). This approach significantly improves GNN performance. However, HL-GNN does not include our domain-specific indices, which limits its performance in certain datasets. Comparative evaluations reveal that PROXI outperforms HL-GNN on OGBL-COLLAB, achieves parity on the PHOTO dataset, and falls short on OGBL-PPA. These results underscore the importance of integrating domain-specific indices into GNN models to optimize performance across diverse datasets.

## C Proximity Indices

In this part, we give the details of our structural and domain indices for each dataset.

**Proximity Indices for All Datasets except OGB datasets.** In all ten homophilic and heterophilic datasets, we used the following indices:

*Structural Proximity Indices.* We have ten structural proximity indices: $\mathcal{J}(u,v), \mathcal{S}_a(u,v), \mathcal{S}_o(u,v), \mathcal{J}^3(u,v), \mathcal{S}_a^3(u,v), \mathcal{S}_o^3(u,v), \mathcal{AA}(u,v), \mathcal{L}_2(u,v), \mathcal{L}_3(u,v)$, and $\mathcal{D}(u,v)$

*Domain Proximity Indices.* We have four domain proximity indices: $\mathcal{CI}(u,v), \mathcal{MC}(u,v), \mathcal{CD}(u,v), \widehat{\mathcal{CD}}(u,v)$

Table 11: Total number of indices used for each dataset in our model.

| | CORA | CITESEER | PUBMED | PHOTO | COMP. | O-COLLAB | O-PPA | TEXAS | WISC. | CHAM. | SQR | CROC. |
|---|---|---|---|---|---|---|---|---|---|---|---|---|
| # Indices | 20 | 19 | 16 | 21 | 23 | 28 | 9 | 18 | 18 | 18 | 18 | 18 |

In particular, in our PROXI, we have 10 structural indices, while the domain indices vary according to the number of classes of each dataset. That is number of domain indices is equal to the number of classes that have fixed dimensions, which is three, added to the dimension of $\mathcal{CI}(u, v)$, which varies according to the number of classes of the dataset. The details of these indices are given in Section 2.2. The importance of each feature for each dataset is given in Table 7.

**Proximity Indices for OGBL-COLLAB.** The OGBL-COLLAB dataset is a time-varying dataset, spanning between years 1963 to the year 2019, where the positive training set spans between years 1963 and 2017, the positive validation set is set the node pairs appearing in the year 2018, and year 2019 is the positive test set. Each link has a weight which is the number of collaborations that occur between the authors pair for the given year. Since it is dynamic and weighted, we needed to adjust our indices in this dataset to adapt our method to this context.

Let $w^y(u, v)$ be the number of collaborations that occur between nodes $u$ and $v$ in the year $y$. For simplicity let us define the total collaborations of a pair $(u, v)$ through all years as

$$\mathcal{W}(u, v) = \{\sum w^y(u, v) : 1963 \leq y \leq 2017\}.$$

We define the number of collaborations of the pair $(u, v)$ between the years 2007 and 2017 as

$$\mathcal{W}_{10}(u, v) = \{\sum w^y(u, v) : 2007 \leq y \leq 2017\},$$

and between years 2012 and 2017 as

$$\mathcal{W}_5(u, v) = \{\sum w^y(u, v) : 2012 \leq y \leq 2017\}.$$

Finally, considering $\mathcal{G} = \{(u, v) : \text{the collaboration between } u \text{ and } v \text{ occurs in years 2007 to 2017}\}$,

$$\mathcal{A}(u) = \{\sum \mathcal{W}_{10}(u, x) : x \in \Gamma(u)\}.$$

$\diamond$ *Author's Oldest Paper Index:* For this feature we track down the year of the earliest paper of each author. If this year is before year 1985, we assign the value 0, and 1 otherwise.

$\diamond$ *Author's Newest Paper Index:* For this feature we track down the year of the latest paper of each author. If this year is before year 1985, we assign the value 0, and 1 otherwise.

$\diamond$ *All Time Collaborations:* For each pair (u,v) we add up the weights of the pair through each year for which the link exists, which is $\mathcal{W}(u, v)$.

$\diamond$ *10-Year Collaborations:* For each pair (u,v) we add up the weights of the pair through each year, from 2007 to 2017, for which the link exists, $\mathcal{W}_{10}(u, v)$.

$\diamond$ *5-Year Collaborations:* For each pair (u,v) we add up the weights of the pair through each year, from 2012 to 2017, for which the link exists $\mathcal{W}_5(u, v)$.

$\diamond$ *All Time Common Collaborators:* For each pair (u,v) we find the neighborhood of node u and node v over the graph created by combining all the years between 1963 and 2017, and we take the intersection of the neighborhoods.

$\diamond$ *10-Year Common Collaborators:* For each pair (u,v) we find the neighborhood of node u and node v over the graph created by combining the years between 2007 and 2017, and we take the intersection of the neighborhoods.

$\diamond$ *5-Year Common Collaborators:* For each pair (u,v) we find the neighborhood of node u and node v over the graph created by combining the years between 2012 and 2017, and we take the intersection of the neighborhoods.

⋄ *Preferential Attachment:* We evaluate Preferential Attachment (Barabâsi et al., 2002) over the graph created by combining the years between 2007 and 2017, which formula is

$$PA(u,v) = \mathcal{A}(u) \cdot \mathcal{A}(v).$$

⋄ *w-Adamic Adar:* Considering the graph $\mathcal{G}$,

$$\mathcal{A}\mathcal{A}^w(u,v) = \sum_{z \in \mathcal{N}(u) \cap \mathcal{N}(v)} \frac{1}{\log |\mathcal{A}(z)|}.$$

⋄ *w-Jaccard Index:* Considering the graph $\mathcal{G}$,

$$\mathcal{J}^w(u,v) = \frac{\{\sum \mathcal{W}_{10}(u,z) + \mathcal{W}_{10}(z,v) : z \in \mathcal{N}(u) \cap \mathcal{N}(v)\}}{\{\sum \mathcal{W}_{10}(u,x) + \mathcal{W}_{10}(x,v) : x \in \mathcal{N}(u) \cup \mathcal{N}(v)\}}.$$

⋄ *w-Salton Index:* Considering the graph $\mathcal{G}$,

$$\mathcal{S}_a^w(u,v) = \frac{\{\sum \mathcal{W}_{10}(u,z) + \mathcal{W}_{10}(z,v) : z \in \mathcal{N}(u) \cap \mathcal{N}(v)\}}{\sqrt{\mathcal{A}(u)\mathcal{A}(v)}}.$$

⋄ *Shortest Path Length:* $\mathcal{D}(u,v)$ over the graph $\mathcal{G}$.

For each node, a 128-dimensional feature vector of word embedings is provided. We set up three proximity to utilize them. These indices are:

⋄ *Common Embedding:* For given word embedding $\mathbf{X}$, we define our *Common Embedding* domain feature $\mathcal{CE}(u,v)$ as the number of matching "1"s in the vectors $\mathbf{X}_u$ and $\mathbf{X}_v$. i.e.,

$$\mathcal{CE}(u,v) = \#\{i \mid \mathbf{X}_u^i = \mathbf{X}_v^i\}$$

⋄ $L^1$ *Distance:* Simply, we use $L^1$-norm (Manhattan metric) in the feature space $\mathbb{R}^m$. If $\mathbf{X}_u = [a_1\ a_2\ \ldots\ a_m]$ and $\mathbf{X}_v = [b_1\ d_2\ \ldots\ b_m]$, we define $\mathfrak{D}(u,v) = \mathbf{d}(\mathbf{X}_u, \mathbf{X}_v) = \sum_{i=1}^m |a_i - b_i|$.

⋄ *Cosine Distance:* Another popular distance formula using some normalization is the cosine distance/similarity. We define our cosine distance feature as

$$\mathfrak{D}^{\mathfrak{c}}(u,v) = \frac{\mathbf{X}_u \cdot \mathbf{X}_v}{\|\mathbf{X}_u\|.\|\mathbf{X}_v\|}$$

⋄ *Year-wise label:* For node pair $(u,v)$, we define $\mathcal{LA}_y = 1$ if the pair exists in year $y$, and 0 otherwise. In our experiments, we iterate $y$ between the years 2007 to 2016.

**Proximity Indices for OGBL-PPA.** OGBL-PPA was a challenging dataset, because of its high number of edges, thus we were limited in using a lot of indices due to memory reasons. For this dataset, we employed 9 proximity indices:

*Structural Indices.* $\mathcal{J}(u,v), \mathcal{S}_a(u,v), \mathcal{S}_o(u,v), \mathcal{A}\mathcal{A}(u,v), \mathcal{L}_2(u,v),$ and $\mathcal{D}(u,v)$

*Domain Indices.* $\widehat{\mathcal{CI}}(u,v), \mathcal{MC}(u,v)$

Taking this into account, in our PROXI for OGBL-PPA we have six structural indices and three domain indices as the *vanilla Class Identifier* $\widehat{\mathcal{CI}}(u,v) = [class(u), class(v)]$ is a 2-dimensional feature. The details of these indices are given in Section 2.2.

## D   More on Transitivity

### D.1   Prediction Heads for Link Prediction

Building on our discussion in Section 3.3, we refer to a recent study that explores the impact of decoders on GNN performance. The authors of Wang et al. (2022a) evaluated various prediction heads and, as previously mentioned, highlighted the poor performance of distance-based (dot-product) decoders due to their inherent limitations in learning efficacy.

Table 12: Performance comparison of different decoders across benchmarks. The results are reported from Wang et al. (2022a, Table 2). Each dataset has its own pre-defined comparison metric indicated below.

| Decoder | Dot Product | Bilinear | ConcatMLP | HadamardMLP |
|---|---|---|---|---|
| **OGBL-DDI (Hits@20)** | | | | |
| GCN [5] | $13.8_{\pm1.8}$ | $16.1_{\pm1.2}$ | $12.9_{\pm1.4}$ | $37.1_{\pm5.1}$ |
| GraphSAGE [12] | $36.5_{\pm2.6}$ | $39.4_{\pm1.7}$ | $34.2_{\pm1.9}$ | $53.9_{\pm4.7}$ |
| Node2Vec [22] | $11.6_{\pm1.9}$ | $13.8_{\pm1.6}$ | $10.8_{\pm1.7}$ | $23.3_{\pm2.1}$ |
| **OGBL-COLLAB (Hits@50)** | | | | |
| GCN [5] | $42.9_{\pm0.7}$ | $43.2_{\pm0.9}$ | $42.3_{\pm1.0}$ | $44.8_{\pm1.1}$ |
| GraphSAGE [12] | $37.3_{\pm0.9}$ | $41.5_{\pm0.8}$ | $37.0_{\pm0.7}$ | $48.1_{\pm0.8}$ |
| Node2Vec [22] | $27.7_{\pm1.1}$ | $31.5_{\pm1.0}$ | $27.2_{\pm0.8}$ | $48.9_{\pm0.5}$ |
| **OGBL-PPA (Hits@100)** | | | | |
| GCN [5] | $5.1_{\pm0.4}$ | $5.8_{\pm0.5}$ | $6.2_{\pm0.6}$ | $18.7_{\pm1.3}$ |
| GraphSAGE [12] | $3.2_{\pm0.3}$ | $6.5_{\pm0.7}$ | $5.8_{\pm0.4}$ | $16.6_{\pm2.4}$ |
| Node2Vec [22] | $4.2_{\pm0.5}$ | $7.8_{\pm0.6}$ | $8.3_{\pm0.4}$ | $22.3_{\pm0.8}$ |
| **OGBL-CITATION2 (MRR)** | | | | |
| GCN [5] | $65.3_{\pm0.4}$ | $69.0_{\pm0.8}$ | $62.7_{\pm0.3}$ | $84.7_{\pm0.2}$ |
| GraphSAGE [12] | $62.2_{\pm0.7}$ | $65.4_{\pm0.9}$ | $60.8_{\pm0.6}$ | $80.4_{\pm0.1}$ |
| Node2Vec [22] | $52.7_{\pm0.8}$ | $54.1_{\pm0.6}$ | $51.4_{\pm0.5}$ | $61.4_{\pm0.1}$ |

### D.2   Transitivity vs. Heterophily

In studying the transitivity behavior of networks, we identify a notable distinction between homophilic and heterophilic graph structures. Homophilic graphs generally exhibit significantly higher transitivity ratios, with the PHOTO dataset being a notable exception. This can be attributed to the strong clustering tendencies in homophilic graphs, where nodes of the same class are more likely to form densely interconnected local communities. These dense clusters positively influence the overall transitivity ratio. Conversely, heterophilic graphs are characterized by a lower likelihood of edges connecting nodes of the same class, leading to reduced clustering and, consequently, lower transitivity. To further illustrate these patterns, we present 2D t-SNE visualizations of NODE2VEC embeddings for both graph types in Figure 4. For these embeddings, the dimensionality was set to 64, with a walk length of 50 and 10 walks per node. These initial findings highlight important structural patterns, but further analysis is needed, which we plan to pursue in future work.

## E   Inductive Performance of PROXI

We present results for the inductive setting on the CORA and TEXAS datasets in Table 13.

In this setting, test edges are sampled exclusively between test nodes, which are removed from the original graph, and the model is trained on the training subgraph. While the performance drop in the homophilic CORA dataset is minimal, the heterophilic TEXAS dataset shows a significant decline, suggesting that structural information alone may be less effective without node features in heterophilic graphs. For reference, Hao et al. (2021) reported AUC scores of 86.4 and 94.5 in inductive and transductive settings

Table 13: AUC results for inductive and transductive link prediction with PROXI.

| Dataset | Inductive | Transductive |
|---|---|---|
| **CORA** | $86.07_{\pm4.59}$ | $95.83_{\pm0.59}$ |
| **TEXAS** | $48.53_{\pm8.22}$ | $84.61_{\pm2.37}$ |

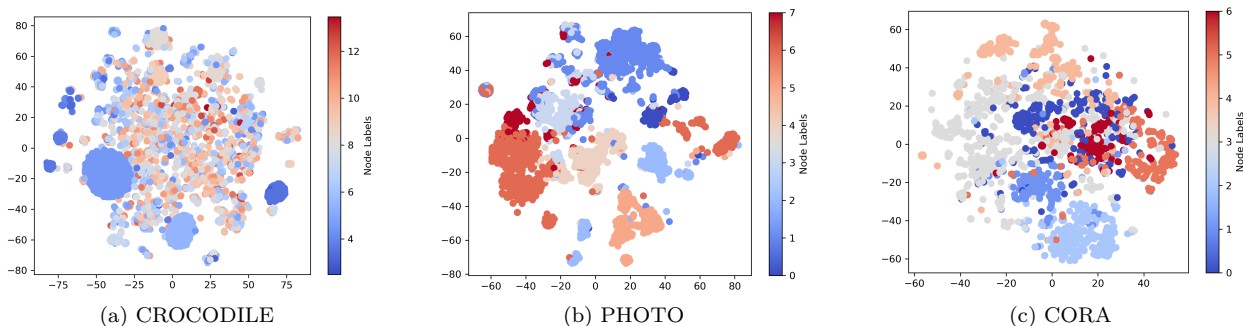

Figure 4: 2D t-SNE visualizations of node2vec vectors of CROCODILE, PHOTO and CORA datasets.

on homophilic datasets, while Naddaf et al. (2023) highlighted
that network density heavily influences generalization from transductive to inductive settings. In future work,
we plan to explore the causes of this performance gap and investigate methods to mitigate information loss.

