# OpenReview forum: "PROXI: Challenging the GNNs for Link Prediction"
_TMLR — Accepted by TMLR_

### Review · Reviewer_h6dX · 2024-12-29

**Summary Of Contributions:**

The paper presents a feature engineering approach (called PROXI) to predict links/edges in a graph. Essentially, the results showcase the benefit of using traditional classification models (such as XGBoost) to leverage structural and domain specific feature information for link prediction tasks. Additionally, these hand-crafted features are incorporated into GNNs to improve their performance.

**Audience:**

Yes

**Claims And Evidence:**

No

**Requested Changes:**

1. Throughout the paper, I couldn't find details about the GNN architectures used as baselines. I checked the code as well but couldn't find any information. More importantly, can the authors add details about the architectures of the GCN, GAT, or SAGE that are used for generating Figure 2?

2. Is it possible to show some results for inductive tasks? In scenarios where node feature information is unavailable, then I think structural information alone on new graphs might not be as useful as expected (when compared to GNNs).

3. **Optional:** Can the authors provide references of previous works that combined feature engineering approaches with GNNs or MLPs for other tasks such as node/graph classification? Did any such approach perform better than GNNs?

4. nit: please adjust the white-space throughout the paper and avoid large empty spaces such as in Page 3.

**Strengths And Weaknesses:**

**Strengths**

1. The paper is mostly well-written and the approach/details are presented clearly.
2. The experiments consider graphs of varying sizes and the link-prediction results are promising when compared to GNN based approaches (especially for heterophilic graphs).

**Weaknesses**

1. Based on Table 8 in the paper, it seems like the 2 hop/3 hop neighborhood information along with the degrees of nodes seem to be the most influential handcrafted features for link-prediction. The paper can be strengthened if a discussion on "why GNNs fail to approximate such 2/3 hop information" can be added. For instance, consider the GCN architecture which relies on the normalized adjacency matrix (say $A$) for neighborhood feature aggregation. By using powers of such a matrix (say $A^2, A^3$), I think the multi-hop structural information can be captured (see [1]).
2. Other than the "graph distance" metric, I believe that the rest of the metrics can be computed with 2 hop/3 hop neighborhood information. More importantly, I looked at the code and seems the authors are relying on the networkx package for graph operations. Based on Table 8, if one were to ignore the "graph distance" metric, then by computing the powers of the adjacency matrix, one can significantly improve the efficiency of the metric calculation functions. This can also be done on a GPU (if available). My concern is that if COMPUTERS dataset (with just 245K edges) takes 18 hours, then it is definitely not a scalable approach. In case, a GPU is not available, I would request the authors to discuss these computational aspects so that future works can address them.
3. In section 3.3, the authors (informally) discuss why cosine similarity of node features might not work for link prediction problems. How does this argument differ for homophilic/heterophilic graphs? In my opinion, this homophily/heterophily nature of the graphs should be considered when presenting the Transitivity ratios in Table 1. Additionally, how do the class labels differ between nodes sharing a common neighbor? Such an analysis can possibly shed light on why PROXI beats the baselines on PHOTO dataset and not on CORA/CITESEER. Let me know if the authors feel otherwise.


**References**

1. Rossi, Ryan A., Nesreen K. Ahmed, and Eunyee Koh. "Higher-order network representation learning." Companion Proceedings of the The Web Conference 2018. 2018.

---

> ### Author Response · Authors · 2024-12-29
>
> Dear Reviewer,
>
> Thank you for your valuable feedback and insightful suggestions. We will address them and follow up with you shortly.
>
> Best regards,
>
> Authors

---

> ### Author Response · Authors · 2025-01-13
>
> *We sincerely thank the reviewer for their valuable feedback, which, together with input from other reviewers, has greatly contributed to improving our paper. The revised version is available via the link provided above. Below, we address your questions and outline the corresponding revisions made to the paper.*
>
> ## W1- GNN &2-3 hop info
> > Based on Table 8 in the paper, it seems like the 2 hop/3 hop neighborhood information along with the degrees of nodes seem to be the most influential handcrafted features for link-prediction. The paper can be strengthened if a discussion on "why GNNs fail to approximate such 2/3 hop information" can be added. For instance, consider the GCN architecture which relies on the normalized adjacency matrix (say A) for neighborhood feature aggregation. By using powers of such a matrix (say A2,A3), I think the multi-hop structural information can be captured (see [1]).
>
> Thank you for this insightful question. We would like to clarify that our claim is not that GNNs are incapable of capturing 2- or 3-hop information, but rather that they do not effectively utilize this information for the link prediction task. Specifically, GNNs primarily learn node embeddings, which are then used for link prediction by adjusting the relative positions of these embeddings for each pair of nodes. While it is possible for GNNs to successfully encode 2- or 3-hop information into node embeddings, this encoding is not effectively utilized for the link prediction task.
>
> This limitation makes GNNs particularly effective for tasks such as node classification, where the objective aligns directly with learning meaningful node embeddings. However, link prediction requires capturing and understanding relationships between pairs of nodes, a need that node embeddings alone do not inherently address. Instead, learning edge embeddings (or node-pair embeddings) is often more appropriate for capturing this relational information. In this context, our PROXI model can essentially be viewed as node-pair embeddings, where each entry corresponds to a specific proximity index.
>
> Our model bridges this gap by explicitly providing the ML model with information about the relative standings of node pairs through a combination of structural and domain-specific indices. During training, the ML model identifies which indices are most relevant to the task and leverages them effectively to enhance performance.
>
> Based on our results, we conclude that while learning node embeddings is highly beneficial for node classification tasks, it may not be as effective for link prediction. For link prediction, approaches that explicitly learn edge embeddings or node-pair embeddings are likely to yield better results.
>
> Thank you for suggesting the use of $A^2$ and $A^3$ to enhance the computational efficiency of our model. We discuss this suggestion in the response to the next question.
>
> ----
>
> ## W2- Computational time improvement
>
> >Other than the "graph distance" metric, I believe that the rest of the metrics can be computed with 2 hop/3 hop neighborhood information. More importantly, I looked at the code and seems the authors are relying on the networkx package for graph operations. Based on Table 8, if one were to ignore the "graph distance" metric, then by computing the powers of the adjacency matrix, one can significantly improve the efficiency of the metric calculation functions. This can also be done on a GPU (if available). My concern is that if COMPUTERS dataset (with just 245K edges) takes 18 hours, then it is definitely not a scalable approach. In case, a GPU is not available, I would request the authors to discuss these computational aspects so that future works can address them.
>
> Thank you for highlighting this issue with our code. We would like to clarify that our goal was to determine the number of simple 2-paths and 3-paths. While $A^2$ can provide this information, $A^3$ does not accurately capture it. Considering this limitation and the size of some datasets (ranging from 10K to 500K nodes), we opted to avoid sparse matrix multiplication and relied on elementary methods instead.
>
> Following your feedback, we revisited our code and realized it could be significantly improved by implementing an optimized function for this task. Previously, our code required 18 hours to process a dataset with 250K nodes. With the optimized function, this time was reduced to less than 15 minutes. In comparison, computing our indices using $A^2$ and $A^3$ took around 90 minutes on the same machine. We updated our code with the new algorithm in the same link, and further details about our runtime are provided in the "Implementation and Runtime" section (Section 4.1).
>
> We sincerely appreciate your suggestion, which has greatly enhanced the computational efficiency and scalability of our model.

---

> ### Author Response · Authors · 2025-01-13
>
> ## W3a - Transitivity vs. Heterophily
>
> >In section 3.3, the authors (informally) discuss why cosine similarity of node features might not work for link prediction problems. How does this argument differ for homophilic/heterophilic graphs? In my opinion, this homophily/heterophily nature of the graphs should be considered when presenting the Transitivity ratios in Table 1.
>
> Thank you very much for this insightful question. For the first part, transitivity with heterophily, we agree with your observation. Transitivity ratios are noticeably higher in homophilic graphs in general with the exception of PHOTO dataset. One explanation for this that in homophilic graphs, the nodes of the same class are more likely to live together and create small clusters. However, in heterophilic graphs, as it is less likely to have an edge between to nodes of the same class, this clustering behaviour does not appear. As they are naturally more inter-connected, these clusters are likely to contribute the transitivity ratio positively. In the figures below, we observe distinct behavior for homophilic and heterophilic datasets.
>
> To illustrate this phenomenon, we included 2D visualizations of homophilic and heterophilic datasets generated using t-SNE method on NODE2VEC embeddings, where node embeddings dimension is set 64, walk length 50 and number of walks 10. In these figures, we observe that homophilic graphs exhibit distinct clustering among nodes, promoting interconnectivity and transitivity. In contrast, heterophilic graphs show weaker clustering and lower transitivity ratios. A short discussion and visualizations are added in Appendix D. However, we believe this insightful question needs more thorough analysis, and we aim to look into this in our future works.
>
> ---
>
> ## W3b - Transitivity vs. Performance
>
> >Additionally, how do the class labels differ between nodes sharing a common neighbor? Such an analysis can possibly shed light on why PROXI beats the baselines on PHOTO dataset and not on CORA/CITESEER. Let me know if the authors feel otherwise.
>
> Thank you for sharing this observation. We calculated the 2-homophily ratios for these three graphs. Specifically, for each dataset, we determined the ratio of node pairs at a distance of 2 that belong to the same class to the total number of such pairs. This is defined as H2=#(cl(u)=cl(v))/∣P∣H_2 = \#(cl(u) = cl(v)) / |P|, where P={(u,v)∣d(u,v)=2}P = \{(u, v) \mid d(u, v) = 2\}.
>
> | Dataset | CORA | CITESEER | PHOTO |
>
> | 2-homophily | 0.8685 | 0.7976 | 0.8302 |
>
> Interestingly, the 2-homophily ratios are even higher than the 1-homophily ratios for these datasets. Notably, the PHOTO dataset exhibits an even higher 2-homophily ratio than CITESEER. This suggests that low transitivity is not the primary reason why PROXI outperforms GNN models, despite GNNs benefiting from transitivity due to their structural properties.
> Furthermore, when examining Table 8, we observe the remarkable effectiveness of the Adamic-Adar index in co-purchasing networks such as PHOTO and COMPUTERS, where its utilization far surpasses that in other datasets. This indicates that specific indices can be highly effective in capturing linking behavior between nodes in certain contexts, while GNNs are unable to inherently capture such index information within their message-passing algorithms.
> The strength of our simple model lies in its versatility. These indices effectively represent the relative positions of node pairs from various perspectives, and the machine learning method can selectively utilize the most relevant indices for downstream tasks.
>
> -----
> -----
>
> ## Requested Changes:
>
> ## R1 - Flowchart for PROXI-GNN
> >Throughout the paper, I couldn't find details about the GNN architectures used as baselines. I checked the code as well but couldn't find any information. More importantly, can the authors add details about the architectures of the GCN, GAT, or SAGE that are used for generating Figure 2?
>
> Thank you very much for this question. We included the flowchart of the model in Appendix B, and give the details in the same section.

---

> ### Author Response · Authors · 2025-01-13
>
> ## R2 - Inductive Tasks
> > Is it possible to show some results for inductive tasks? In scenarios where node feature information is unavailable, then I think structural information alone on new graphs might not be as useful as expected (when compared to GNNs).
>
> Thank you for your question. We present our results for the inductive setting on the CORA and TEXAS datasets. In this setting, all test edges are selected between test nodes, which are removed from the original graph, and all training is conducted on the training subgraph. Interestingly, while the information loss in the homophilic setting is relatively low, the link prediction performance decreases significantly in the heterophilic setting. In our follow-up project, we plan to investigate the reasons behind this performance drop and explore alternative methods to mitigate the information loss.
>
> For reference, we reviewed the literature and found that Hao et al. [1] evaluated both inductive and transductive settings for homophilic datasets, achieving AUC scores of 86.4 and 94.5 for CORA for the inductive and transductive settings, respectively. Furthermore, a recent study [2] explored the generalizability of models from transductive to inductive settings, highlighting that performance is highly influenced by the density of the network. We added these results and short discussion the inductive performance of our model in Appendix E.
>
> | Dataset | CORA | TEXAS |
> |---------|---------------|----------------|
> | Inductive| 86.07 ± 4.59 | 48.53 ± 8.22 |
> | Transductive | 95.83±0.59 | 84.61±2.37 |
>
> [1] Hao, Y., Cao, X., Fang, Y., Xie, X. and Wang, S., Inductive link prediction for nodes having only attribute information. In IJCAI 2021.
>
> [2] Naddaf, P., Mahmoudzaheh Ahmadi Nejad, E., Zahirnia, K., Jaeger, M. and Schulte, O., 2023, October. Joint Link Prediction Via Inference from a Model. In Proceedings of the 32nd ACM International Conference on Information and Knowledge Management (pp. 1877-1886).
>
>
> ----
>
> ## R3 - Other tasks for Feature Engineering &GNN
>
> >Optional: Can the authors provide references of previous works that combined feature engineering approaches with GNNs or MLPs for other tasks such as node/graph classification? Did any such approach perform better than GNNs?
>
> We added the following recent references at the end of Section 2.1, where topological and traditional feature engineering methods outperform or give competitive results with GNNs in graph classification and node classification tasks.
>
> **Graph Classification:**
>
> Chen, Y., et al., EMP: Effective Multidimensional Persistence for Graph Representation Learning. In Learning on Graphs Conference 2024.
>
> Loiseaux, D. et al. A framework for fast and stable representations of multiparameter persistent homology decompositions. NeurIPS 2024
>
> **Node Classification:**
>
> Uddin, M.J., et al.,. ClassContrast: Bridging the Spatial and Contextual Gaps for Node Representations. arXiv preprint arXiv:2410.02158.
>
> Luan, S., et al. When do we need graph neural networks for node classification?. In International Conference on Complex Networks and Their Applications (2023). Cham: Springer Nature Switzerland

---

### Review · Reviewer_5U5K · 2024-12-30

**Summary Of Contributions:**

The authors challenge the current status quo of using Graph Neural Networks (GNNs) for link prediction. They present an alternative method, PROXI, which makes use of proximity information of node pairs, using both structural information and node attributes. They leverage several proximity-based methods to compute a set of features (or indices) and train a classifier model to predict links between node pairs. The authors provide a comprehensive summary of their method, clearly state their assumptions and compare the performance of their model to a multitude of baselines. Remarkably, they show that their method is not only competitive but outperforms the current state of the art on multiple benchmarks. Secondly, they show that PROXI can be combined with existing GNN architectures, resulting in significant performance improvements. Finally, the authors perform ablation studies to better understand the importance of different classes of proximity indices. Overall, the paper successfully challenges the over-usage of GNNs for link prediction tasks and shows that a traditional feature-engineering approach can indeed outperform the current state of the art.

**Audience:**

Yes

**Claims And Evidence:**

Yes

**Requested Changes:**

1. If possible, although I understand the space limitations, I would like to see an impact analysis on the adaptability of PROXI to other settings, where the current set of indices may not be applicable.
2. Please try to improve the clarity of the paper by using less technical concepts or giving a short introduction to them the first time they appear.
3. Section 3.3. needs some references, to make it more concrete and not overly speculative.
4. Include a discussion/ justification for using different baselines in Tables 3 and 4.
5. Please make sure to name the performance metric in Table 4 as well (I suppose it is AUC but I saw no mention of this).

**Strengths And Weaknesses:**

### Strengths
1. Refreshing - the authors take inspiration from traditional approaches and show that applying deep neural networks is not always the best approach.
2. Comprehensive - the authors present their method to a sufficient degree clearly and concisely.
3. Well-motivated - the authors clearly present the motivation for their study and also cite recent work on existing issues with GNNs.
### Weaknesses
1. Adaptability - as the authors also point out, their method is difficult to adapt to new domains, which is a significant limitation compared to GNN-based approaches.
2. Clarity - the paper's first two sections heavily rely on the difference between homophily and heterophily. An unfamiliar reader has to get to section 3.2 to get an understanding, but many paragraphs before it already assume familiarity with them.
3. Section 3.3 - The speculative analysis in this section (and also in the introduction) lacks any references to existing models that use such distance metrics as prediction heads. It would be nice to see a performance comparison between "similarity-based" and learned (e.g. MLP) prediction heads in existing GNN models.
4. Tables 3 and 4 - I noticed the reported performances in Table 4 do not appear in Table 3 (except for LINKX) and the baselines in Table 4 also have no citations. Is there any particular reason for this?

---

> ### Author Response · Authors · 2024-12-31
>
> Dear Reviewer,
>
> Thank you very much for your feedback and suggestions. We greatly appreciate your input and will follow up soon.
>
> Best regards,
> The Authors

---

> ### Author Response · Authors · 2025-01-13
>
> *We sincerely thank the reviewer for their valuable feedback, which, together with input from other reviewers, has greatly contributed to improving our paper. The revised version is available via the link provided above. Below, we address your questions and outline the corresponding revisions made to the paper.*
>
> ## W1 - Adaptability
>
> > Adaptability - as the authors also point out, their method is difficult to adapt to new domains, which is a significant limitation compared to GNN-based approaches.
>
> Thank you for raising this concern. We respectfully disagree with the reviewer’s assertion, as our computationally efficient model is highly adaptable to various domains and graph sizes. This adaptability stems from the fact that our structural indices are computed directly from the adjacency matrix, making them applicable to any graph. While domain-specific indices depend on the format of the feature vectors, our model effectively handles diverse types, including binary, real-valued, and other commonly used formats in benchmark datasets. This flexibility underscores the robustness and general applicability of our approach. Our experiments further confirm this point, as the benchmark datasets we used span multiple domains, graph sizes, and diverse characteristics.
>
> We would like to emphasize that our goal in this paper is not to propose a new model to replace GNNs but to demonstrate that current GNN models are not substantially superior to traditional methods. Additionally, we aim to show how traditional methods can be effectively leveraged to enhance the performance of GNNs for link prediction task.
>
> ---
>
> ## W2 - Clarity
>
> > Clarity - the paper's first two sections heavily rely on the difference between homophily and heterophily. An unfamiliar reader has to get to section 3.2 to get an understanding, but many paragraphs before it already assume familiarity with them.
>
> Thank you for the helpful suggestion. We have added a brief discussion of these concepts in the introduction to make the paper more accessible to a general audience. After the rebuttal period, we plan to thoroughly review the paper and further improve the exposition while adhering to the page limits.
>
> ---
>
> ## W3 - Prediction Head Effect
>
> > Section 3.3 - The speculative analysis in this section (and also in the introduction) lacks any references to existing models that use such distance metrics as prediction heads. It would be nice to see a performance comparison between "similarity-based" and learned (e.g. MLP) prediction heads in existing GNN models.
>
> Thank you for raising this concern. Similar issues with GNNs have also been noted in recent studies  [1,2], which we have now cited in our discussion in Section 3.3. For a more detailed comparison of different decoders in link prediction, we refer to a recent study [3] that thoroughly evaluates the most commonly used prediction heads on benchmark datasets. These references have been added to Section 3.3. We added the performance table from [3]  to our appendix (Table 12).
>
> [1] Zhu, J.,et al. Pitfalls in link prediction with graph neural networks: Understanding the impact of target-link inclusion & better practices. In Proceedings of the 17th ACM International Conference on Web Search and Data Mining (2024).
>
> [2] Cho, Y.S., Decoupled Variational Graph Autoencoder for Link Prediction. In Proceedings of the ACM on Web Conference 2024.
>
> [3] Wang, Y. et al,. Flashlight: Scalable link prediction with effective decoders. In Learning on Graphs Conference (2022).
>
> ---
>
> ## W4 - Performance Tables
> > Tables 3 and 4 - I noticed the reported performances in Table 4 do not appear in Table 3 (except for LINKX) and the baselines in Table 4 also have no citations. Is there any particular reason for this?
>
> Thank you for the opportunity to clarify. As stated in the caption,  in Table 3, we adopted the baselines reported in the recent work by Zhou et al. (2022) and followed the same experimental settings. In Table 4, we conducted experiments using the same settings for foundational GNN models—GCN, GAT, and GSAGE—to highlight the performance improvements achieved by PROXI in these key architectures, which serve as the basis for modern GNN models. Similarly, in Table 5, we reported the baselines from Li et al. (2023), and in Table 9, we presented the baseline performances from Zhao et al. (2022). This approach allowed us to compare PROXI with a wide range of recent models, which is why not all models across the performance tables are identical.

---

> ### Author Response · Authors · 2025-01-13
>
> ## Requested Changes:
>
> ## R1 - Adaptability Experiments
>
> > If possible, although I understand the space limitations, I would like to see an impact analysis on the adaptability of PROXI to other settings, where the current set of indices may not be applicable.
>
> Please refer to our response for Weakness 1 above. We also added a short discussion and results for the inductive performance in Appendix E.
>
> ------
>
> ## R2 - Clarity
>
> > Please try to improve the clarity of the paper by using less technical concepts or giving a short introduction to them the first time they appear.
>
> Thank you very much for this suggestion. We added description of homophily and heterophily to introduction. After the rebuttal period, we plan to thoroughly review the paper and further improve the exposition while adhering to the page limits.
>
> --------
>
> ## R3 - Prediction Head references
>
> > Section 3.3. needs some references, to make it more concrete and not overly speculative.
>
> Thank you for this suggestion. We added several references to Section 3.3 emphasizing the decoder effect on GNN performance for link prediction task, and added the performance comparison table for these decoders from one of these references to Appendix D.
>
> ------------
>
> ## R4 - Performance Tables
>
> > Include a discussion/ justification for using different baselines in Tables 3 and 4.
>
> Thank you for sharing this concern. We have added a sentence in Section 4.2 to clarify that the baseline results in the tables are directly sourced from the references cited in the table captions. The variations in baseline results arise from differences in experimental settings and benchmark datasets used in these references. To ensure a comprehensive and fair comparison, we opted to use the reported performances from these recent and relevant studies.
>
> ----------
>
> ## R5 - minor
>
> >Please make sure to name the performance metric in Table 4 as well (I suppose it is AUC but I saw no mention of this).
>
> Thanks for catching this. We revised the caption in Table 4.

---

### Review · Reviewer_izxa · 2025-01-07

**Summary Of Contributions:**

This work explores the effect of augmenting modern machine learning methods with structural features (eg # of common neighbors). In doing so, the authors introduce PROXI, which is a structural feature augmentation for use in standard link-prediction settings that captures higher-order structure features. The authors carry out empirical experiments to justify their methodological claims, and show that PROXI provides some utility when used in conjunction with traditional backbones.

**Audience:**

Yes

**Claims And Evidence:**

No

**Requested Changes:**

Please see the weaknesses section

**Strengths And Weaknesses:**

**Strengths:**
1. The work is presented in a clear and easy to understand fashion
2. The authors present experiments that are easy to understand and provide justification for many of their claims.
3. The authors provide clear and intuitive arguments for the failing of learned structural node representations for link prediction. Insights like these are rarely shared in the literature, and I appreciate them.

**Weaknesses:**
1. The presented method seems like it might struggle with computational complexity. Could you detail the computational complexity for your PROXI scores? And is it possible to use techniques like those outlined in Chamberlin et al. to reduce this computational cost? Additionally, could you comment on whether this technique is suitable for situations where you have dynamic negative resampling?
2. What is the operative difference between PROXI (or at least, the structural components of it) and BUDDY/ELPH, aside from the latter method using hashing based methods to approximate these neighborhood indices. It would seem to me that your $\mathcal{L}_3$ index would be expressible through a mix of their approximate overlap operators.
3. It seems as if PROXI seems to struggle in multiple settings -- could you comment as to why this might be the case?
4. In what ways is PROXI similar to a labeling trick? Given that labeling tricks can also fed into a GNN in ways similar to PROXI, it is worth adding them as a baseline. Specifically double radius node labeling (DRNL) might be valuable and related.
5. It seems as if you're missing other heuristic-learning baselines such as HLGNN [1] that might be relevant.
6. The argument in 3.3 feels strange. Triangle closures are a common heuristic that performs quite well for link prediction, and has powered many methodological improvements [2]. Beyond this, the argument that link predictors that _only_ take in structural node representations can lead to degenerate learned representations is not new and has been proven rigorously [3] and described intuitively [4].
7. The related work section is missing many relevant contributions. For structural indices it's missing for example the local path index [5], extensions of it [6], and the Katz index [7], among others. For structurally augmented GNN based link predictors the authors are missing references to SEAL, ELPH/BUDDY, NeoGNN, and HLGNN among others. Because of this, I would recommend that the authors give a more thorough review of the literature to help further situate the work.

[1] https://arxiv.org/abs/2406.07979

[2] https://dl.acm.org/doi/10.1145/3459637.3481920

[3] https://arxiv.org/abs/1910.00452

[4] https://proceedings.neurips.cc/paper_files/paper/2021/file/4be49c79f233b4f4070794825c323733-Paper.pdf

[5] https://journals.aps.org/pre/abstract/10.1103/PhysRevE.80.046122

[6] https://www.nature.com/articles/s41598-020-76860-2

[7] https://link.springer.com/article/10.1007/bf02289026

---

> ### Author Response · Authors · 2025-01-07
>
> Dear Reviewer,
>
> Thank you for your valuable feedback and thoughtful suggestions. We are addressing them and will follow up with you soon.
>
> Best regards,
> The Authors

---

> ### Author Response · Authors · 2025-01-13
>
> *We sincerely thank the reviewer for their valuable feedback, which, together with input from other reviewers, has greatly contributed to improving our paper. The revised version is available via the link provided above. Below, we address your questions and outline the corresponding revisions made to the paper.*
>
> ## W1 - Computational Complexity
>
> > The presented method seems like it might struggle with computational complexity. Could you detail the computational complexity for your PROXI scores? And is it possible to use techniques like those outlined in Chamberlin et al. to reduce this computational cost?
>
> Thank you for your question. The computational complexities of our similarity indices are $\mathcal{O}(|V| \cdot k^3)$, where $|V|$ represents the total number of nodes, and $k$ is the maximum degree in the network. Following Reviewer h6dX's suggestion, we revisited our code and made significant improvements to the runtime of our experiments. Initially, we relied on existing methods to compute these indices; however, by incorporating recent references, as you suggested, we achieved substantial enhancements in runtime efficiency. For example, the end-to-end runtime (including the computation of proximity indices and the machine learning classifier) for OGBL-COLLAB is now 30 minutes. Similarly, for COMPUTERS (250K nodes), the runtime for all indices is 15 minutes.
> Our structural indices are indeed similar to those introduced by Chamberlain et al., and their code could potentially be applied to our methods. However, given the computational efficiency of our current implementation, we have not evaluated the suitability of their code. Further details about our runtime are provided in the "Implementation and Runtime" section (Section 4.1).
>
> ---
>
> ## W1b- Dynamic Negative Resampling
>
> > Additionally, could you comment on whether this technique is suitable for situations where you have dynamic negative resampling?
>
> Thank you for your question. PROXI is suitable for dynamic negative resampling if it leverages real-time structural and domain proximities, allowing it to adapt to changing negative samples. However, computing and maintaining a pool of hard negatives during training can be computationally costly. Aside from this, our simple design can integrate effectively with dynamic sampling methods and should give competitive performance across various datasets and settings.
>
> ----

---

> ### Author Response · Authors · 2025-01-13
>
> ## W2- Similarity with BUDDY
>
> > What is the operative difference between PROXI (or at least, the structural components of it) and BUDDY/ELPH, aside from the latter method using hashing based methods to approximate these neighborhood indices. It would seem to me that your L3 index would be expressible through a mix of their approximate overlap operators.
>
> Thank you for this insightful question. The BUDDY/ELPH paper, as noted in their introduction, was also motivated by the observed poor performance of GNNs on link prediction tasks. They specifically highlight the challenges in transitioning from node representations to link predictions, which they identify as a key reason for the suboptimal performance. To address this issue, they propose enhancing MPNNs by incorporating subgraph information into the message-passing algorithm, leveraging the structure of node neighborhoods.
> While this approach, through hashing, implicitly captures many of the structural indices used in our model, it does not incorporate our domain-specific indices. These indices play a crucial role in significantly improving our model’s performance, as shown in Table 8. Moreover, while our model is not inherently learnable, it provides a more direct and comprehensive way to inform the ML model about the relationships between node pairs. The ML model dynamically selects which indices to utilize based on the dataset, offering remarkable versatility and adaptability to our approach. In this context, we observe that our model outperforms BUDDY in both OGBL-COLLAB and OGBL-PPA datasets (table below).
>
> That said, how our indices and domain-specific information can be effectively integrated into modern GNNs remains an open question. Although BUDDY/ELPH attempts to capture such information via subgraph hashing, the core issue persists: adapting the link prediction problem by learning node representations. While we show the effectiveness of these indices when they are directly incorporated into prediction heads for node pairs (Table 4), we believe this is not the optimal way to leverage these indices.
>
> In our ongoing work, we are developing a new GNN model specifically designed for link prediction tasks. This model focuses on learning edge (or node pair) representations, using our indices as initial messages. Our goal is to compare the performance of GNNs that learn edge representations versus those that learn node representations for link prediction tasks. Through this comparison, we aim to demonstrate that a "one-size-fits-all" approach is inadequate for GNNs in this context. We added this comparison and short discussion of these models in Appendix B.1.
>
>
> ---------------------------------------------------------------
> | Models        | OGBL-COLLAB          | OGBL-PPA             |
>
> |     ----                | Hits@50   | AUC      | Hits@100  | AUC      |
>
> ---------------------------------------------------------------
>
> | BUDDY         | 64.59 ± 0.46 | 96.52 ± 0.40 | 47.33 ± 1.96 | 99.56 ± 0.02 |
>
> | PROXI         | **76.50 ± 0.27** | **97.24 ± 0.03** | **50.36 ± 0.76** | **99.90 ± 0.00** |
>
> ---------------------------------------------------------------
>
> ----
>
> ## W3 - Adaptability
>
> > It seems as if PROXI seems to struggle in multiple settings -- could you comment as to why this might be the case?
>
> Thank you for raising this concern. We respectfully disagree with the reviewer’s assertion and would like to clarify that our computationally feasible model is indeed highly adaptable to various domains and graph sizes. Unlike GNNs, our model is not inherently learnable. However, the way our indices comprehensively inform the ML model about the relative positions of node pairs—both structurally and within a domain-specific context—makes it remarkably versatile and adaptable.
>
> A key strength of our approach is that the structural indices can be efficiently computed for any graph, as they depend solely on the adjacency matrix. While the domain-specific indices are influenced by the format of the feature vectors, our model’s ability to handle a wide variety of feature types—such as binary, real-valued, and others commonly found in benchmark datasets—further underscores its adaptability to diverse settings. This flexibility highlights the robustness of our method. Our experiments further confirm this point, as the benchmark datasets we used span multiple domains, graph sizes, and diverse characteristics.
>
> We would also like to emphasize that the purpose of this paper is not to propose a new model to replace GNNs. Instead, our goal is to demonstrate that current GNN models are not substantially superior to traditional methods and to show how these traditional approaches can be effectively leveraged to enhance GNN performance for link prediction tasks.

---

> ### Author Response · Authors · 2025-01-13
>
> ## W4 - Labeling Trick vs. PROXI
>
> > In what ways is PROXI similar to a labeling trick? Given that labeling tricks can also fed into a GNN in ways similar to PROXI, it is worth adding them as a baseline. Specifically double radius node labeling (DRNL) might be valuable and related.
>
> Thank you for introducing this highly relevant method into the discussion. Similar to BUDDY, the Labeling Trick paper by Zhang et al. (2021) addresses the challenges associated with generating meaningful representations for node sets from individual node representations, highlighting the limitations of such approaches in their study.
>
> While BUDDY effectively captures most of our structural indices, the labeling trick focuses on certain domain-specific indices we use. For instance, the zero-one labeling directly aligns with our “class identifier” and “common class” indices by using node classes as labels. Additionally, our domain indices, such as common digits and normalized digits, could potentially be adapted within the labeling trick framework, though this would require further work.
>
> It is worth noting that our PROXI model outperforms all three GNN models enhanced with the labeling trick, as reported in Zhang et al. (2021), on both the OGBL-PPA and OGBL-COLLAB datasets (table below). We have included the table and a discussion of these models in Appendix B.1.
>
> -----------------------------------------------
>
> | Model     | OGBL-PPA (Hits@50) | OGBL-COLLAB (Hits@100) |
>
> -----------------------------------------------
> | BUDDY     | 47.33 ± 1.96       | 64.59 ± 0.46          |
>
> | GCN+DE    | 36.48 ± 3.78       | 64.44 ± 0.29          |
>
> | GCN+DRNL  | 45.24 ± 3.95       | 64.40 ± 0.45          |
>
> | SEAL      | 48.80 ± 3.16       | 64.74 ± 0.43          |
>
> -----------------------------------------------
>
> | PROXI     | **50.36 ± 0.76**   | **76.50 ± 0.27**      |
>
> -----------------------------------------------
>
>
> ## W5 - Similarity with HLGNN
>
> > It seems as if you're missing other heuristic-learning baselines such as HLGNN [1] that might be relevant.
>
> Thank you very much bringing this recent publication to our attention. Indeed, HL-GNN successfully integrates several structural indices we use (Table 2 in HL-GNN paper) effectively to their message aggregation scheme, and hence they significantly improves the performance of GNN models. However, they fail to incorporate our domain indices to their algorithm. When we compare our performance with Hl-GNN, we outperform them in OGBL-COLLAB, and have the same result in PHOTO datasets. HL-GNN outperform PROXI in OGBL-PPA. We added a discussion of this work in Appendix B along with BUDDY and Labeling Trick models.
>
> ----
>
> ## W6 - Transitive Prediction Heads
>
> > The argument in 3.3 feels strange. Triangle closures are a common heuristic that performs quite well for link prediction, and has powered many methodological improvements [2]. Beyond this, the argument that link predictors that only take in structural node representations can lead to degenerate learned representations is not new and has been proven rigorously [3] and described intuitively [4].
>
> Thank you for raising this concern. Similar issues with GNNs have also been noted in recent studies, which we have now cited in our discussion [1,2]. For a more detailed comparison of different decoders in link prediction, we refer to a recent study [3] that thoroughly evaluates the most commonly used prediction heads on benchmark datasets. These references have been added to Section 3.3. We added the performance table from [3] to our appendix (Table 12), which shows that distance-based decoders perform poorly compared to other prediction heads.
>
> [1] Zhu, J.,et al. Pitfalls in link prediction with graph neural networks: Understanding the impact of target-link inclusion & better practices. In Proceedings of the 17th ACM International Conference on Web Search and Data Mining (2024).
>
> [2] Cho, Y.S., Decoupled Variational Graph Autoencoder for Link Prediction. In Proceedings of the ACM on Web Conference 2024.
>
> [3] Wang, Y. et al,. Flashlight: Scalable link prediction with effective decoders. In Learning on Graphs Conference (2022).
>
> ---
>
> ##  W7 - Additional Related Work
>
> > The related work section is missing many relevant contributions. For structural indices it's missing for example the local path index [5], extensions of it [6], and the Katz index [7], among others. For structurally augmented GNN based link predictors the authors are missing references to SEAL, ELPH/BUDDY, NeoGNN, and HLGNN among others. Because of this, I would recommend that the authors give a more thorough review of the literature to help further situate the work.
>
> Thank you for bringing these valuable references to our attention. We have incorporated all the suggested works into appropriate sections of the paper. The baseline references are given in the accuracy tables.

---

### Decision · Action_Editor_wLRY · 2025-02-05

**Recommendation:** Accept as is

**Comment:**

After the rebuttal period, all of the reviewers unanimously recommend acceptance of this work to TMLR. I concur with the reviewers, and believe this will be a valuable addition to the link prediction literature. Well done!

**Audience:**

This work presents a strong baseline for link prediction which is competitive against graph neural networks. This will be a very valuable starting point for link prediction evaluation, and is very likely to attract a sizable audience in this space.

**Claims And Evidence:**

The empirical evaluation of the paper is well-executed and supports the key claims of the paper, meeting the bar for TMLR.